# Models transport Saharan dust too low in the atmosphere: a comparison of the MetUM and CAMS forecasts with observations

Debbie O'Sullivan[1], Franco Marenco[1], Claire L. Ryder[2], Yaswant Pradhan[1], Zak Kipling[3], Ben Johnson[1], Angela Benedetti[3], Melissa Brooks[1], Matthew McGill[4], John Yorks[4], Patrick Selmer[4]

[1]Met Office, Exeter, EX1 3PB, UK

[2]Department of Meteorology, University of Reading, RG6 6BB, UK

[3]European Centre for Medium-Range Weather Forecasts, Reading, RG2 9AX, UK

[4]NASA Goddard Space Flight Center, Greenbelt, MD 20771, USA

*Correspondence to*: Franco Marenco (franco.marenco@metoffice.gov.uk)

**Abstract.** We investigate the dust forecasts from two operational global atmospheric models in comparison with in-situ and remote sensing measurements obtained during the AERosol properties – Dust (AER-D) field campaign. Airborne elastic backscatter lidar measurements were performed on-board the Facility for Airborne Atmospheric Measurements during August 2015 over the Eastern Atlantic, and they permitted to characterize the dust vertical distribution in detail, offering insights on transport from the Sahara. They were complemented with airborne in-situ measurements of dust size-distribution and optical properties, and datasets from the Cloud-Aerosol Transport System spaceborne lidar (CATS) and the Moderate Resolution Imaging Spectroradiometer (MODIS). We compare the airborne and spaceborne datasets to operational predictions obtained from the Met Office Unified Model (MetUM) and the Copernicus Atmosphere Monitoring Service (CAMS). The dust aerosol optical depth predictions from the models are generally in agreement with the observations, but display a low bias. However, the predicted vertical distribution places the dust lower in the atmosphere than highlighted in our observations. This is particularly noticeable for the MetUM, which does not transport coarse dust high enough in the atmosphere, nor far enough away from source. We also found that both model forecasts underpredict coarse mode dust, and at times overpredict fine mode dust, but as they are fine-tuned to represent the observed optical depth, the fine mode is set to compensate for the underestimation of the coarse mode. As aerosol-cloud interactions are dependent on particle numbers rather than on the optical properties, this behaviour is likely to affect their correct representation. This leads us to propose an augmentation of the set of aerosol observations available on a global scale for constraining models, with a better focus on the vertical distribution and on the particle size-distribution. Mineral dust is a major component of the climate system, therefore it is important to work towards improving how models reproduce its properties and transport mechanisms.

## 1 Introduction

Mineral dust is an important component of the Earth system (Forster et al., 2007, Haywood and Boucher, 2010, Knippertz and Todd, 2012), and it affects the scattering and absorption of solar and infrared radiation, as well as cloud microphysics. The Saharan desert is the main source of mineral dust (Washington et al., 2003; Shao et al., 2011), and once lifted into the air the dust can be transported over thousands of kilometres (Knippertz and Todd,

2012., Tsamalis et al., 2013) where it is exposed to the effects of ageing and mixing. These effects change its optical, microphysical and cloud condensation properties (Richardson et al., 2007., Lavaysse et al., 2011), affecting the size distribution, the chemical composition, and the radiative effects. The transported dust also affects tropical cyclone development through effects on the sea surface temperature (Evan et al., 2018), and the deposition of iron-rich material into the ocean has an impact on biogeochemical cycles (Jickells et al., 2005).

Dust is forecast prognostically in Numerical Weather Predictions (NWP) because of its impacts on atmospheric circulation (Solomos et al., 2011; Mulcahy et al., 2014), visibility, air quality, health and aviation. Significant progress has been made in dust modelling over the last decade, with a suite of regional and global dust models now available. In recent years dust models have also started to assimilate aerosol optical depth (AOD) measurements from satellites (Niu et al., 2008; Benedetti et al, 2009; Liu et al., 2011; Di Tomaso et al, 2017). There have been a number of studies in recent years to provide further insight on the transport and properties of dust (e.g. Heintzenberg, 2009, Ansmann et al., 2011, Kanitz et al., 2014, Ryder et al., 2015, Groß et al., 2015 among many more), and the ability of models to predict dust events (e.g. Chouza et al., 2016, Ansmann et al., 2017.) However, there have been few studies assessing how well the vertical distribution of dust is captured in models. For example, Chouza et al., (2016) found that the ECMWF MACC model (precursor to the CAMS model considered here) simulated Saharan plumes that matched the vertical distribution, but underestimated the marine boundary layer aerosol extinction, compensating the missing AOD with an overestimate of the dust layer intensity. More recently, Ansmann et al., (2017) found that dust models, including the one run at ECMWF, were able to forecast dust well for the first few days after emission, but that the modelled loss processes were too strong, leading to an underestimation with increasing distance from source. Other studies have shown that dust is not optimally represented in models, highlighting insufficient uplift and insufficient transport of the coarser particles. For example, Evan (2018) found that the representation of dust in climate models was affected by errors in the surface wind fields over Northern Africa. Given the diversity of findings and the range of available models and methodologies, there is a continued need to assess the model predictions of the dust vertical distribution, particularly with information on vertically resolved particle size information, which is not usually available from operational remote sensing observations.

Aerosol Robotic Network (AERONET) sun photometer retrievals (Holben et al., 1998) play an important role in dust model evaluation (for example see: Scanza et al., 2015; Cuevas et al., 2015; Ridley et al, 2016) and offer near continuous measurements and, for some stations, long observation records. However, AERONET instruments do not provide information on vertical distribution. Dry convective mixing can raise mineral dust to altitudes of at least 5-6 km over the Sahara, and disperse it into a deep mixed layer (Messager et al., 2010). The dominant Easterly winds at these latitudes advect this airmass across the Atlantic Ocean, and as the hot, dry and dust-laden air passes the West African coast, it is undercut by cooler moist air in the marine boundary layer (MBL) and forms an elevated layer called the Saharan Air Layer (SAL) (Karyampudi et al., 1999). As plumes move across the Atlantic, the altitude of the SAL may decline due to large-scale subsidence and loss processes, and the residence time of the lofted dust is closely related to the height and size distributions. High-latitude dust lifted in Iceland during winter storms has also been reported up to to high altitude, with coarse particles up to 5 km (Dagsson-Waldhauserova, 2019). The impact of dust on radiation and clouds also depend on its vertical distribution (Johnson et al., 2008).

The key loss processes, wet and dry deposition and turbulent downward mixing, are strongly influenced by the altitude of the dust and the fine and coarse mode fractions. Note that in this paper we will denote particles with diameters < 1 μm as fine mode dust, with coarse mode particles having diameters >1 μm.

Lidar observations provide valuable information about the location and vertical distribution of aerosols in the atmosphere, and can as such be useful in the evaluation dust models. Spaceborne lidar measurements provide this information on a global scale. For example, the Cloud-Aerosol Lidar with Orthogonal Polarization (CALIOP) on board the Cloud-Aerosol Lidar and Infrared Pathfinder Satellite Observations (CALIPSO) is an elastic backscatter lidar system (Winker et al., 2010) with limited capability to distinguish different types of aerosol (Omar et al., 2009). The Cloud-Aerosol Transport System (CATS) onboard the International Space Station was a polarisation sensitive backscatter lidar with good detection sensitivity and ability to differentiate different aerosol types (Yorks et al., 2016). Both systems include depolarisation measurements, which permits the identification of mineral dust reliably vs other aerosol types. Airborne lidar measurements of aerosols typically offer a finer resolution and the combination with a number of other airborne instruments, but on a limited geographical scale (see e.g. Marenco et al, 2011; Marenco, 2013; Marenco et al, 2016).

In this work we compare airborne measurements of mineral dust with model predictions. The measurements include remote sensing with elastic-backscatter lidar and in-situ dust observations of the particle size distribution. We also make use of data from the CATS spaceborne lidar to extend our analysis over the Sahara. The observations are used to assess the performance of the dust forecast from two operational global models, the Met Office Unified Model (MetUM) and the European Centre for Medium-range Weather Forecasts, Copernicus Atmosphere Monitoring Service (ECMWF-CAMS) model. The data are used to investigate whether convection, largescale wind, boundary layer height, or dust size distribution have the greatest effect on how well the models capture the vertical structure of the dust layers.

## 2 Models

In this study observation data is used to assess the relative performance of the dust schemes in two operational global models. Both models and their respective dust schemes are briefly described in section 2.1 and 2.2. Both models considered here assimilate MODIS AOD into the model analysis to improve the AOD forecast (e.g Pope et al., 2016), and the models perform generally well for the prediction of dust AOD. For this study, short range forecasts were used (forecast lead time < 12 hours).

### 2.1 MetUM

The Met Office Unified Model (MetUM) is a non-hydrostatic, fully compressible, deep-atmosphere dynamical core, solved with a semi-implicit semi-Lagrangian time step on a regular latitude-longitude grid (Davis et al., 2005). The configuration used in this study is the Global NWP model that was operational in 2015 (Global Atmosphere 6.1), which had a resolution of 0.35° longitude by 0.23° latitude, corresponding to an approximate resolution of 25 km at mid-latitudes and ~40 km at the equator (Walters et al, 2017). There are 70 vertical levels, reaching an altitude

of 80 km (Pope et al., 2016). The dust scheme uses 9 size bins for the horizontal flux calculations with diameters between 0.0632 μm to 2000 μm, and either a 6 or 2 bin scheme for the subsequent transport and advection (Woodward 2001; Woodward 2011; Collins et al., 2011; Brooks et al, 2011). The operational Global model, used here, uses the 2-bin dust scheme: division 1 (d1) covers the 0.2 - 4.0 μm diameter range, and division 2 (d2) the 4.0 - 20 μm diameter range).

AOD from MODIS collection 5.1 on-board the Aqua satellite was assimilated into the model, from Deep Blue over land, and Dark Target over selected ocean regions in the dust belt (note that ocean assimilation was at that time limited to grid points with observed AOD > 0.1). There are four daily model runs, initialised at 00, 06, 12 and 18 UTC, and the model fields are available with a timestep of 3 hours (00, 03, 06, 09, 12, 15, 18 and 21 UTC). See Pope et al., (2016) and references therein for a description of how the model is initialised and the AOD data assimilation methodology.

The extinction efficiency for each of the MetUM dust bins is pre-calculated into a lookup-table, based on Mie scattering with an assumed underlying log-normal distribution and the refractive index from Balkanski et al (2007). The extinction coefficient is then determined in the model, by multiplying the predicted mass mixing ratio by the pre-computed extinction efficiency (Johnson and Osborne, 2011).

## 2.2 ECMWF – CAMS

The global atmospheric composition forecasts run at ECMWF, as part of the Copernicus Atmospheric Monitoring Service (CAMS), are a continuation of the work of the Monitoring Atmospheric Composition and Climate (MACC) project. The CAMS system combines state-of-the-art modelling with Earth observation data assimilated from a variety of sources, including MODIS Collection 5 AOD from Aqua and Terra (limited to Dark Target retrievals). The data used here are from the operational forecasts produced in near real time during the period of the ICE-D campaign. At that time, the horizontal resolution was 80km (corresponding to a T255 spectral truncation) and there were 60 vertical levels. The model provided a 120 h long forecast from 00UTC, and the analysis used 12-hourly 4D-VAR data assimilation, using MODIS Terra and Aqua Dark Target AOD to constrain the total aerosol mixing ratio. Details of the model set up and the analyses can be found in Morcrette et al., (2009), Benedetti et al., (2009), and Cuevas et al., (2015). The operational CAMS global assimilation and forecasting system uses fully integrated chemistry in the ECMWF Integrated Forecasting System (IFS), for this time period cycle 40r2. The IFS is a spectral model using vorticity-divergence formulation with semi-Lagrangian advection and physical parameterizations on a reduced Gaussian grid. The CAMS aerosol parameterization is based on the LOA/LMD-Z (Laboratoire d-Optique Atmosphérique/Laboratoire de Météorologie Dynamique-Zoom) model (Reddy et al., 2005). Prognostic aerosol of natural origin such as mineral dust and sea salt are described using three size bins. In total CAMS has 5 different types of prognostic aerosol, unlike the MetUM which only has dust in the operational model. For dust the bin size classes are one fine mode (division 1 or d1, 0.06-1.1 μm diameter), and two coarse mode bins (division 2 or d2: 1.1-1.8 μm diameter, and division 3 or d3: 1.8-40 μm diameter). Morcrette et al., (2009) state that the size bins are chosen such that the mass concentration percentages are 10% for the fine dust mode, and 20% and 70% for the two coarse dust size bins during emission.

Extinction coefficient is computed in the model for each aerosol bin, by multiplying the mixing ratio by the mass extinction coefficient derived from offline Mie scattering calculations based on the optical properties of Dubovik et al. (2002) as documented in Morcrette et al. (2009). For dust, hygroscopic growth is not considered.

## 3 Measurements and instrumentation

### 3.1 ICE-D campaign

AERosol properties – Dust (AER-D) was a collaborative campaign led by the Met Office in collaboration with the Universities of Reading and Hertfordshire (Marenco et al, 2018). It was held at the same time as the Ice in Clouds Experiment – Dust (ICE-D), a larger collaborative campaign involving the Met Office, the, National Centre for Atmospheric Science (NCAS), Manchester and Leeds Universities (UK), the British Antarctic Survey and Mainz University. In addition, the Sunphotometer Airborne Validation Experiment in Dust (SAVEX-D) was also carried

out, thanks to EUFAR funding based on a proposal from the University of Valencia, Spain, the Met Office and the University of Reading. SAVEX-D is treated here as a component of AER-D. The AER-D and ICE-D field campaigns were conducted on 6-25 August 2015 from Praia, Cape Verde (14°57'N, 23°29'W), 650 km off the West coast of Africa, an ideal region for observing dust outflow. The main aim of the ICE-D campaign was to characterise the properties of Saharan dust as ice nuclei (IN) and cloud condensation nuclei (CCN), their impact on

cloud microphysical processes, and the formation of convective and stratiform clouds. The AER-D and SAVEX-D projects aimed at characterizing dust properties above the Eastern Atlantic. The main measurements were made using the Facility for Atmospheric Airborne Measurements (FAAM) Airborne Research Aircraft, a modified BAe-146-301, and in total, 16 flights took place between both campaigns, six of which contained high-altitude sections dedicated to surveying the vertical distribution of dust using lidar. The instruments deployed on the aircraft enabled

a range of measurements of aerosol size distribution, chemical composition, optical properties and radiative effects. Most flights took place in proximity of the Cape Verde Islands, with the exception of flights B923, B924 and B932, which sampled between Cape Verde and the Canaries. Ground-based measurements were also made on the island of Santiago during the month. These experiments together provide a comprehensive dataset to investigate the properties of transported Saharan dust during the summer season. The key airborne instruments, and satellite data

used in this study are briefly discussed in the next sections.

    During AER-D and ICE-D, Saharan air masses were transported by predominantly Easterly winds over the Atlantic in a sequence of events between the 6th and 25th August. Cape Verde was often on the edge of the transported dust, enabling flights to sample the main dust plume and a gradient across the flight track. The dust episodes often lasted for several days, which provided the opportunity to make measurements of dust of varying age. Among the

key aims of the AER-D project are the improvement of dust remote sensing from space and from the ground, and the validation of dust predictions in the MetUM and other models. The focus of the present paper is on the latter objective. Four dust events are considered here, derived from five research flights (one event having been sampled through a double flight). A summary of the flight sections considered is given in Tables 1 and 2, and the flight tracks are shown in Fig 1. We use these data to investigate whether convection, largescale wind, boundary layer

height, or dust size distribution have the greatest effect on how well the models capture the vertical structure of the

dust layers. There is no direct measure for convection in the archived model fields, as such the impact of convection on the dust forecast can only be inferred through a process of elimination.

For convenience flight sections are divided in "runs" and "profiles": we have a run (also called straight and level run, and denoted here with the letter R) when the aircraft flies for a certain time on a constant heading and a constant altitude, and a profile (denoted here with the letter P) when the aircraft changes altitude with a constant rate of ascent or descent. Note that an aircraft profile is a slant trajectory through the atmosphere and thus differs from a lidar profile (vertical). Each aircraft run or profile is identified with a number, hence for a given flight we have R1, R2, … and P1, P2, …. The runs and profiles of interest in this paper are identified in Tables 1 and 2.

### 3.2 Airborne lidar

The Leosphere ALS450 elastic backscatter lidar (wavelength 355 nm) is deployed on the FAAM aircraft in a nadir-viewing geometry. Marenco et al., 2011, and Marenco, 2013 describe the methodology for converting lidar beam returns at 355 nm wavelength into profiles of aerosol extinction coefficient. The system specifications are summarised in Marenco et al., 2014 and references therein, and a further description of the data processing methodology can be found in Marenco et al., 2016. During processing, the lidar data was integrated to 1 min temporal resolution, which corresponds to a $9 \pm 2$ km footprint at typical aircraft speeds. Smoothing to a 45 m vertical resolution was also applied to reduce the effect of shot noise. The vertical profiles were processed using a double iteration. First we determined the lidar ratio (extinction-to-backscatter ratio), and subsequently we processed the full data set to determine the extinction coefficient and AOD (see Marenco et al., 2016 and references therein, where the same methodology is applied). The first iteration was conducted on a subset of the vertical profiles, where the signature of Rayleigh scattering above the dust layer could clearly be identified to enable the lidar ratio to be determined. We obtained a campaign mean lidar ratio of $54 \pm 8$ sr, which is in reasonable agreement with other measurements of the lidar ratio for dust at 355 nm (Lopes et al., 2013). This value of the lidar ratio was subsequently used to process the full dataset in the second iteration. On average during this campaign, the uncertainty in the derived dust extinction coefficient was 8%; however, with a significant variability of this figure in both the vertical and horizontal. The uncertainty is smaller than this near the top of the profile (closer to the aircraft) and larger nearer the ground. The methodology described in Marenco et al., (2016) was used here.

### 3.3 In-situ aerosol measurements

A number of wing-mounted instruments permitted us to measure the aerosol size distribution between 0.1 and 100 µm. The Passive Cavity Aerosol Spectrometer Probe (PCASP; Liu et al., 1992; Osborne et al., 2008; Rosenberg et al., 2012) measured optical size from 0.1-2.5µm. The cloud droplet probe (CDP-100; Lance et al., 2010; Rosenberg et al., 2012) measured larger particles with diameters 5-40 µm (Knollenberg, 1981), and the two-dimensional stereo probe (2DS) measured large aerosol particles up to ~100 µm. Calibration of the PCASP was done before and after the campaign, whereas the CDP was also calibrated before most flights. The PCASP and CDP measurements (d <20 µm) and their calibration for the ICE-D campaign are discussed in more detail in Ryder et al (2018), where the full size distribution measurements are described. The particle size spectra have been processed for an assumed refractive index for dust of 1.53−0.001i, thus correcting for the bin ranges calibrated using polystyrene latex

spheres, and the first bin has been discarded due to its undefined lower edge. The 2DS is a shadowing probe with 10 µm resolution, and it does not rely on refractive index to infer particle size. Profiles of in situ measurements were acquired on slant trajectories through the atmosphere (aircraft profiles).

## 3.4 Satellite datasets

Two sources of satellite data are used here, the Cloud-Aerosol Transport System (CATS) and the Moderate Resolution Imaging Spectroradiometer (MODIS). CATS is a multi-wavelength lidar instrument (wavelengths 532 and 1064 nm) developed to enhance Earth Science remote sensing capabilities from the International Space Station (ISS) (McGill et al., 2015). CATS operated for 33 months (10 February 2015 to 29 October 2017), primarily in an operating mode that was limited to the 1064 nm wavelength due to issues with stabilizing the frequency of laser 2 (Yorks et a., 2016). The CATS Level 1 data product includes 1064 nm attenuated total backscatter (ATB) and linear volume depolarization ratio measurements. Yorks et al. (2016) provides an overview of the CATS L1 data products and processing algorithms and a comparison with airborne data. Pauly et al. (2019) found that the CATS 1064 nm ATB has a low bias of up to 7% in aerosol layers compared to airborne and ground based lidars due primarily to CATS calibration uncertainties. The CATS extinction coefficient profiles have a 5 km horizontal resolution (along-track) and 60 m vertical resolution. Lee et al. (2019) showed that CATS extinction profiles compared favorably with CALIPSO, with differences due to the aforementioned ATB bias and differences in parameterized extinction-to-backscatter ratios. This paper utilizes the vertical profiles of 1064 nm aerosol extinction coefficient in the CATS Level 2 (L2) Version 3-01 5km Profile products derived from the L1 attenuated total backscatter data. For this study, the data were filtered by the 'cloud' and 'invalid' flags, thus showing only the aerosol data points. The aerosol subtype (plotted together with extinction coefficient) indicates that most of the aerosol of interest here is in fact classified as dust and dust mixtures in the CATS L2 dataset.

MODIS collection 6.1 level-2 atmospheric aerosol product from Aqua (MYD04_L2) and Terra (MOD04_L2) were obtained from the Level-1 and Atmosphere Archive & Distribution System (LAADS, ftp://ladsftp.nascom.nasa.gov/allData/61/). The merged Deep Blue and Dark Target aerosol optical depth at 550 nm from both Aqua and Terra was used to create daily AOD maps (Hsu et al, 2004, 2006, 2013; Levy et al, 2013; Sayer et al, 2013, 2014). The differences between the collection 5 (used in both models for operational assimilation in August 2015) and the subsequently released collection 6 is treated in detail in the above referenced papers. Generally speaking, with the collection 6 update, the Deep Blue product was extended to vegetated surfaces, and improvements to the aerosol type classification and quality assurance were introduced for both the Dark Target and Deep Blue products. Comparisons performed by the authors suggest that, generally speaking, the collection 6 AOD values are marginally higher in the dust source regions (e.g. Western Africa and Middle East). The differences between the MODIS collections represent a major improvement to the MODIS product, but we do not expect them to substantially affect the conclusions drawn in this paper.

## 3.5 Analysis of dust source regions and transport

Source regions for the sampled dust were investigated using two back-trajectory models. Back-trajectories were calculated from the time, latitude, longitude and altitude of various points along the flight track where high dust

loadings had been encountered using the Numerical Atmospheric Modelling Environment (NAME) (Jones et al, 2007) and the Hybrid Single-Particle Lagrangian Integrated Trajectory model (HYSPLIT) (Draxler and Hess, 1998; Stein et al, 2015). In the NAME back-trajectories, meteorological data from MetUM was used, and the HYSPLIT back-trajectories were driven using meteorological data from the National Oceanic and Atmospheric Administration (NOAA) Global Data Assimilation System (GDAS). Despite the very different models and meteorological data used, the back-trajectories from the two models highlighted consistent source regions.

Haboobs driven by convective outflows from mesoscale storms have been shown to represent the dominant uplift mechanism of Saharan dust during the summer months, with a share of 50% of the uplifted dust (Marsham et al., 2013a). The meteorological reanalyses driving HYSPLIT back-trajectories and NAME dosage maps are not able to identify the dust source location, or potentially the transport pathways over these or subsequent uplift events (Sodemann et al., 2015). On the other hand, haboobs and dust storms are clearly identified by an expert eye in the EUMETSAT "dust RGB" product from the MSG/SEVIRI infrared channels (http://oiswww.eumetsat.int/~idds/html/product_description.html), and it is thus possible to utilise this type of imagery to track dust as it is transported, thus helping to determine source location and uplift time (e.g Schepanski et al., 2007). Dust events observed during four of the flights considered here were examined in this way by Ryder et al. (2018) and mesoscale convective storms drove the dust uplift and subsequent transport in all of them. Despite the inability of back trajectory analysis to really capture haboobs, the back-trajectories and the satellite-tracking of the plumes gave consistent results.

The identified source regions and dust transport paths are shown in Fig 1. This uses a combination of work done by Ryder et al. (2018) and Liu et al. (2018), with additional information in this work from NAME and HYSPLIT to help identify the dust trajectory. A detailed discussion on the meteorology during the ICE-D campaign can be found in Liu et al, (2018). A key point is that the MBL in the Eastern Atlantic was typically 300 – 500 m deep during the study period, which is in agreement with the aircraft lidar observations and the in-situ measurements during aircraft ascent and descent profiles. Liu et al., (2018) also show that on the 15th of August there was a change in the synoptic conditions. This means that for the first and third case study used here (B920 and B927) the maximum horizontal wind speed above the MBL, in the Saharan Air Layer (SAL) was lower than 10 m/s and wind direction varied between NE and SE which resulted in lower dust loadings during these two flights. Case study 2 (B923 and B924) was also in this period of slower windspeeds, but high dust loadings were sampled due to the more northerly location of flights B923 and B924. In the final case study looked at here, case study 4 (B932), the wind speed above 2km was significantly enhanced with a more Easterly wind direction. This resulted in higher dust loadings being observed in case study 4 than for 1 or 3 – note that the highest dust loadings of all were observed in case study 2 due to the location of these flights, see Liu et al., (2018) for the full meteorological and dust source analysis.

## 3.6 Comparison of datasets

The airborne lidar measurements of aerosol extinction coefficient and AOD were measured at a wavelength of 355 nm, whereas the MODIS and AERONET data used here were all collected at 550 nm, and CATS aerosol properties are at 1064 nm. The model extinction is available for a variety of wavelengths including 380, 550 and 1064 nm,

and for CAMS 355 nm is also available. Here, the MetUM dust aerosol extinction coefficient was re-calculated from the mass concentrations of division 1 and division 2 dust (see section 2.1 for a description of the dust scheme), and Mie-derived optical properties of the two dust size bins.

Having measurements at different wavelengths across datasets hasn't been a major concern, because very little wavelength dependence was noted during the campaign for aerosol extinction: the difference in AOD between 340 and 550 nm was less than 5% in the Aeronet data examined. Similarly, the MetUM extinction at 355 nm was only $22 \pm 7\%$ larger than at 1064 nm. This is explained with the small Ångström exponent during the campaign (-0.4 to 0.4: see Liu et al., 2018), and this is generally expected for coarse mineral dust particles. For this reason, it was deemed unnecessary to scale extinction and AOD for wavelength in the present study.

The MetUM extinction coefficient only includes dust, which could potentially make the results lower compared to total aerosol extinction which also includes other aerosol types. However, data from the CATS lidar, as well as the in-situ measurements including filter samples discussed in Ryder et al (2018), confirm that the aerosol sampled during AER-D/ICE-D was predominantly dust, with a contribution from marine aerosol in the MBL. For this reason, for this study we neglect the conceptual difference between the dust-only extinction of the MetUM, and total-aerosol properties in CAMS and the observations.

The comparison methodology used is summarised in Fig. 2.

## 4 Results and discussion

In sections 4.1 and 4.2, the measurements of aerosol extinction coefficient, AOD and dust concentration for the different size bins used by the MetUM and CAMS are used to assess the predicted dust, as well as the representation of dust size distribution in both models. In sections 4.3 and 4.4, the model largescale wind and boundary layer height are compared with observations to infer what if any influence these have on the dust forecast.

### 4.1 Individual case studies

**Case study 1: 7th August 2015, B920 (Fig 3 - 8).** This flight took place near Praia and was co-located with an overpass of the CATS spaceborne lidar. There were two high level sections during the flight that have been looked at, R1 and R6 (see Table 1 for run times and locations). Fig 3 displays the airborne, spaceborne and model data for R6, which coincided with a CATS overpass. A deep dust layer was observed between ~ 2 and 5 km, with marine aerosol mixed with dust in the boundary layer, and a broken cloud field at the top of the boundary layer. Both the extent and amount of aerosol observed agree well between the airborne and the spaceborne lidars (Fig 3 a and d). The aerosol type classification from CATS (not shown here) also agrees well with the in-situ measurements, which found a marine aerosol layer below the dust layer. The dust layer was well mixed, with moderate extinction coefficients (100 - 180 Mm$^{-1}$) and AOD's between 0.28 and 0.44 observed by the airborne lidar.

Fig 3e-g show that the models and the observations display a low AOD around Cape Verde, with much larger values near the Canary Islands and off the West African coast. In Fig. 3g, the AOD observations from MODIS, AERONET (stars) and the aircraft lidar (dots) are in agreement within 5%. This broad agreement is consistent with the fact that both models assimilate MODIS AOD. However, the MetUM and CAMS models underpredict the intensity of the AOD maximum by 0.9 and 0.6, respectively, and there are also variations in the predicted plume location.

From Fig 3a-d we see that the predicted vertical distribution of the dust layer shows some differences from the observations: the dust layer extends from the surface to around 4 km in the MetUM and from 1 to ~ 4 km in CAMS, whereas CATS and the airborne lidar both show the dust layer between 2 and 5 km. The magnitude of the extinction coefficient predicted by the models of $100 - 170$ $Mm^{-1}$ is however in good agreement with the observations from both lidars ($100$-$200$ $Mm^{-1}$). The mean, standard deviation and maximum extinction values for each considered flight section are summarised in Table 1. For this run, the MetUM mean extinction was $55 \pm 38$ $Mm^{-1}$, ECMWF forecast $58 \pm 41$ $Mm^{-1}$ and the aircraft lidar measured a mean extinction value of $56 \pm 40$ $Mm^{-1}$.

In Fig 4a the mean extinction profile for R6 is shown for the airborne lidar, the MetUM and the CAMS model, and Fig 4b displays the mean dust concentration profile in each of the size bins for both models for the same time period. As already highlighted from Fig 3 the MetUM has the dust layer extending right down to the ocean surface. It is moreover dominated by the smaller size bin (d1, $0.2 - 4.0$ μm diameter), in particular for the aerosol below 1km primarily. The concentration predicted by CAMS for this case is about half of that in the MetUM, and however the magnitude of the predicted extinction is similar, around $100$-$120$ $Mm^{-1}$. There are, however, differences in the dust layering, for the MetUM the maximum is near the surface with a smooth decline with altitude, whereas CAMS predicts an elevated dust layer between 1 and 4 km as discussed for Fig. 3. This discrepancy in concentrations is thought to be mainly ascribed to the representation of the particle size-distributions, whereas the agreement in terms of extinction can be understood if one considers that the models are tuned to the observations.

The dust concentration from the MetUM divisions d1 and d2 and the CAMS divisions d1 (0.06-1.1 μm diameter), d2 (1.1-1.8 μm diameter), and d3 (1.8-40 μm diameter) have also been compared with the in-situ measurements for each of the 5 size ranges, as well as the total dust concentration measured during aircraft profiles. Two profiles from this flight are shown in Fig 5 and 6. The observed concentration of dust in the MetUM d1 size bin typically makes up about a third of the total dust concentration measured, and d2 is around two thirds. In contrast, the measurements only show $0 - 10$ μm $m^{-3}$ dust in the CAMS d1 and d2 size bins, and the concentration in the d3 size bin is very close to the total measured. Comparing the model data (lines with markers on) to the measurements (lines of the same colour with no markers) in Fig 5 and 6 we can see that both models struggle to accurately capture dust concentration for each size bin. This adds to the difficulty in attributing dust to the right altitude. For example, in P2 (Fig 6a) the MetUM has more d1 dust than d2, while the aircraft measurements show the opposite. For the same profile (Fig 6b), CAMS has more d2 dust than d3, however the measurements show that there is less than 10 μm $m^{-3}$ d1 or d2 dust, and the predicted CAMS d3 shows a maximum of 60 μm $m^{-3}$, to be compared with 350 μm $m^{-3}$ (observed maximum).

Temperature and specific humidity profiles from the aircraft in-situ instruments were also compared with data from the MetUM and ECMWF. An example is shown for this flight for P2 (Fig 7) and P7 (Fig 8). The temperature profiles are within 3.5 degrees in the boundary layer and within 1.5 degrees above 4.5 km, with no systematic bias for either model. Both models also generally get the specific humidity profiles about right, capturing the main features, although with more obvious differences than for temperature. Generally, the models predict a correct vertical structure of the atmosphere in terms of thermodynamic profiles; however the predicted dust vertical distribution seems to depart excessively from the thermodynamic structure.

**Case study 2: 12[th] August 2015, B923 and B924 (Fig 9-11).** Flights B923 and B924 both took place on the 12[th] August flying between Praia and Fuerteventura to sample the outflow from a dust uplift event that had happened on the 10[th] August in Northern Mali. These flights were able to reach the main dust plume, which means that highest AOD's and extinction coefficients of the campaign were measured on this day (Marenco et al, 2018). The two flights sampled the same plume at different times during the day, and only B923 is shown here as results for flight B924 are similar. The AOD measured by the airborne lidar reached 2, with an aerosol extinction coefficient of $100 - 1300$ Mm$^{-1}$ near the Western African coast. As in the previous case study, both models captured the spatial distribution of the dust AOD well (Fig 9d-f); however the MetUM underpredicted the intensity of the AOD maximum by 1.1, and the CAMS model underpredicted it by 0.8.

For this section of flight B923, both models showed a dust layer up to $\sim 5$ km, with an enhanced extinction coefficient at 13-17°W, between the surface and 1 km, where the extinction coefficient increases from an average in-layer value of 100-150 Mm$^{-1}$ to 500-700 Mm$^{-1}$ (Fig 9a-b and Fig 10. This spatial distribution along the flight track is similar to the observed one (Fig 9c and 10); however, the maximum dust extinction is observed at $\sim 1$ km altitude, whereas the models predict it closer to the surface, and the dust maximum extinction coefficient along the flight track was under-predicted in the MetUM and CAMS by 45% and 80%, respectively (Table 1). Two sections of flight B924, on the same day, reinforce these results (not shown here as they are similar to the section just discussed). However, Fig 9d-f shows that there is a difference in the general representation by both models: CAMS predicts a maximum AOD of 1.6, with almost the same values and spatial distribution that were observed by lidar, whereas the MetUM underpredicts this dust event's maximum AOD by 0.6 compared to the lidar, and 1.5 compared to MODIS. The differences between models and observations could possibly be associated to the dust having been uplifted by a strong haboob, for which models, running with the resolution and convection parameterisation required for global coverage, are unlikely to represent in a way that gives the strength of the uplift (Marsham et al. 2013b, Birch et al. 2014, Roberts et al., 2018). In particular, we note that the convection parametrisation has no specific representation of surface gusts due to downdrafts (main contributors to dust uplift) and that it isn't currently coupled to the dust scheme.

In P1 the measurements show very large amounts of dust, up to 3000 $\mu g/m^{-3}$ concentration (Fig 11), with both models predicting significantly less (250 $\mu g/m^{-3}$ in MetUM, and 120 $\mu g/m^{-3}$ in CAMS). Interestingly in this aircraft profile, which is closer to the area affected by the intense dust, both models have the more dust in the largest size bins, in agreement with the in situ measurements.

In summary, compared to the very large differences between the measured and modelled dust concentration, the modelled extinction is much closer to the observations.

**Case study 3: 15th August, B927 (Fig 12-13).** This case study is quite interesting, as the dust was confined to a shallow layer between 2.0 and 3.5 km as can be seen in fig 12c. The extinction coefficient (~100 - 300 Mm[-1]) measured by the lidar, and the AOD (up to 0.36) were moderate. Much higher AOD values, up to 2.4, were observed by MODIS over Africa and nearer to the coast. As can be seen from Fig 12a-c, the ECMWF CAMS model does a good job at getting the dust layer centred around an altitude of 2.9 km and with an extinction coefficient of 180-330 Mm[-1], in good agreement with the observations, but with a larger layer depth (between 1.5 and 4 km). This is particularly noticeable in the run mean plot (Fig 13a). On the other hand, the MetUM predicts a dust layer centred around 2.7 km, close to the lidar observations, but the peak extinction coefficient is under-predicted by ~200 Mm[-1]. A second dust layer is predicted near the surface below 1.1 km, and this results in an AOD range of 0.4 – 1.8, which is similar to the AOD range of 0.3 – 2.0 predicted by CAMS (Fig 12d+e). The location of the maximum AOD predicted by the models is in reasonable agreement with the MODIS observations (Fig 12f), however MODIS observed higher AOD values in the dust plume than the models predicted, up to 2.6.

Fig 13b shows the modelled dust mass concentrations in the different size bins. For the MetUM there is a greater amount of dust in the smaller size bin, with a peak in d1 dust of $120 \pm 10$ μg/m$^{-3}$ and a peak in d2 dust of $70 \pm 20$ μg/m$^{-3}$. For the CAMS model the opposite is true and smallest size bin peaks at $20 \pm 7$ μg/m$^{-3}$ in the main dust layer, with most of the dust mass in the larger two size bins reaching a maximum of $100 \pm 9$μg/m$^{-3}$ for d2, and 80 $\pm$ 7 μg/m$^{-3}$ for d2.

**Case study 4: 20th August, B932 (Fig 14 – 16).** The fourth case study shows another interesting flight, where the dust was observed in an elevated layer between 2 and 4.5 km (Fig 14c). For the dust observed on this day, the estimated transport time from the source region was 2.5 days, thus shorter compared to the previous three. The dust was uplifted by a mesoscale convective system on the 17th August near the Algeria/Mali border and from the northernmost tip of Mali (Fig 1). The aerosol extinction coefficient (~100 and 400 Mm[-1]) and AOD observed by the airborne lidar (up to 0.72) were the highest observed during the campaign after B923 and B924. We note that this flight also travelled about ~800 km to the Northeast of Cape Verde, hence getting closer to the main plume. As can be seen from Fig 14d-f, the AOD in the dust plume is between 0.6 and 1.2 for both models, which compares well to the 0.7 – 1.4 observed by MODIS. Both models simulate the spatial distribution of the AOD well compared to observations and predict the observed North-South gradient along the flight track. From Fig 14a-c we can also see that both models forecast the top of the dust layer reaching around 4 km, which is only slightly lower than the 4.5 km observed on the lidar. However, the observations show most of the dust in a relatively shallow layer between 2 and 3.5 km, whereas the models have the peak of the dust below 1 km. This can also be seen quite clearly in Fig 15a.

Out of the 8 high level sections from the 4 case studies included in this work, R1 from B932 shown here is the only case study where both models predict a higher extinction coefficient than was observed by airborne lidar. As can be seen from Table 1, the lidar measured a mean aerosol extinction coefficient of $76 \pm 81$ (Max 395 Mm[-1]), while the MetUM and ECMWF mean and max values were $140 \pm 130$ (Max 620 Mm[-1]) and $140 \pm 120$ (Max 500 Mm[-1]),

respectively. In this case, moreover, both models have most of the dust concentration in the largest size bin, although the d2 (4.0 – 20 μm) dust mass for the MetUM is underestimated by 20% and the CAMS d3 (1.8 – 40 μm) dust mass is underestimated by 85% compared with observations. The peak d2 mass of 800 ± 200 μg/m$^{-3}$ predicted by the MetUM is 270 μg/m$^{-3}$ larger than the peak d1 mass of 520 ± 90 μg/m$^{-3}$. (Fig 15b). In CAMS, the peak d2 and d3 mass in the dust layer are ~200 ± 75 μg/m-3 each, i.e. more than double of the peak d1 mass of 90 ± 10 μg/m-3. Still, the fine mode dust appears overestimated by ~30% , ~80%, and ~90% for the MetUM d1, and the CAMS d1 and d2, respectively (peak model value compared to peak observed). The greater contribution of the smaller dust particles to the extinction coefficient, combined with overestimation of the overall concentration are consistent with the predicted extinction coefficient being ~12% and ~40% higher than the observed one for this case study, for CAMS and the MetUM, respectively. Note that the CAMS d2 dust mass concentration of R1 (Fig 15b) and P4 (Fig 16b) is virtually identical to the d3 mass concentration, with the two lines overlapping.

## 4.2 General findings from the four case studies considered

For all the case studies the MetUM and ECMWF global dust forecasts capture the spatial distribution of dust AOD reasonably well in comparison with observations. The model predictions show some positioning errors compared to MODIS AOD, and this can affect the local comparisons made at the aircraft location. In the case studies considered, the models showed underprediction of the AOD by 0.8–1.5 and 0.6–0.9 for the MetUM and CAMS, respectively. However, in case study 4 both models underpredicted the AOD by ~0.2.

The model prediction of the vertical distribution of the dust extinction coefficient is not always consistent with observations. As a general rule, we have observed that both models have tended to predict the dust 0.5 – 2.5 km too low in the atmosphere, compared with the observations, with ECMWF generally better capturing elevated dust layers. The ECMWF CAMS model also captures the depth of the dust layer better than the MetUM, with the height of the dust layer being more accurate and with the MetUM often extending the dust layer down to the surface, in cases when this is not seen in the observations. In the next section we will use data from the CATS spaceborne lidar, in comparison with predictions from the MetUM, to investigate what could be causing the observed discrepancies in the dust vertical distribution.

We noted large differences of 25 – 100% (corresponding to 100 – 2800 μg/m-3) between the measured and modelled dust concentration, associated however with a modelled extinction within ~50% of the observations, which may appear surprising because concentration is the modelled variable, from which optical properties are computed. We need to bear in mind, however, that AOD is the mostly used metric used to compare aerosol model predictions and observations: AERONET AOD is often used in model verification, and both the MetUM and the CAMS model use MODIS AOD in data assimilation. It is not so surprising, therefore, that modelled optical properties are pulled towards the observations, even when the microphysical properties from which they are computed are out of scale (in this case, an underestimated dust concentration). Finer particles make a greater contribution to the aerosol extinction coefficient per unit mass than coarser ones, and the mismatch between the representation in concentration and in optical properties can be compensated in the models through the size-distribution. For most of the aircraft profiles studied here, the models have about a factor of 2 too much dust in the

smaller size bins, meaning that an underpredicted dust concentration can yield an aerosol extinction coefficient of the right order of magnitude.

For the flights which sampled dust nearer the source regions (case studies 2 and 4) the models had 65–90% of the dust concentration in larger size bins (MetUM d2 and CAMS d3) than for the other flights, where this proportion was 35–60%. This seems to indicate that the models may represent the dust size distribution better nearer the source. The observations from the AER-D and ICE-D campaigns suggest that, as the dust travels away, the observed size distribution changes little, with large particles transported in significant quantities as far as Cape Verde (Liu

et al., 2018; Ryder et al., 2018). In contrast, the models appear to lose particles from the larger size bins rapidly with increasing dust mass age, due to the gravitational sedimentation processes.

**4.3. Comparison with the CATS spaceborne lidar**

We compared almost every CATS overpass covering North Africa and the Eastern Atlantic during AER-D and ICE-D with the MetUM. CATS and model data were compared for overpasses between 6 and 25 August 2015, in

the study region off the Western African coast between 40°N and 10°S latitude and 40°W and 40°E longitude, for a total of 45 overpasses. The four most significant cases are discussed here. For each overpass, the CATS aerosol extinction coefficient was compared with the MetUM dust extinction coefficient, and the modelled contribution to extinction of each of the two size bins was also analysed.

In fig 17, a CATS overpass at 00 UTC on the 7[th] August over the African continent is shown, with significant

amounts of dust between 1 and 7 km. The MetUM predicts the dust in more or less the right places across the CATS track, but underpredicts the magnitude of the extinction coefficient by 60%. As for the case studies in section 4.1, most of the predicted dust is also lower in altitude than in the observations (below 5km) than observed and extends to the surface (although the model does predict some dust reaching as high as 7 km). The smaller size bin contributes 80% of the modelled extinction coefficient.

In fig 18 a CATS overpass from 18 UTC also on the 7[th] August is shown, where the dust is moving off from the West African coast over the sea. At the Eastern end of the transect the model has a similar dust extinction coefficient (60-180 Mm$^{-1}$) to CATS (80-260 Mm$^{-1}$), the key difference being that the model layer extends between the surface and 5 km, while in the CATS observations it extends between 1 and 7 km. However, over the ocean (longitude > 15°W) the model misses the layer evident in the CATS data.

Two further examples shown in fig 19 for 00 UTC on the 8[th] August and fig 20 for 16 UTC on the 10[th] August, showing a similar pattern. In fig 19 the entire CATS overpass shown is over land: at the Northwest end of the overpass both the MetUM and CATS show the dust plume extending from the surface to over 7km. However, towards the Southeast the model predicts it to be between the surface and ~ 4 – 6 km, whereas CATS continues showing the layer between 1 and 7 km. The model predicts ~65% less extinction coefficient than CATS.

In Fig 20, similar to Fig. 18 the CATS overpass starts over the West African coast and then moves over the ocean. As in the previous example, the model predicts a deep dust layer extending up to 6km. The model underpredicts the aerosol extinction by ~ 65%, and by ~45% over land. Over land, division 2 predicted dust makes up 7.5% of

the dust concentration, dropping away to nearly zero over the ocean, potentially due to sedimentation of the coarser particles.

Two things stand out from the above examples: (1) over the African continent, where the dust is uplifted, the model generally agrees better with the observations than over the ocean further away from the source region, and (2) the smaller dust particles (division d1) in the model reach the same altitude as the dust layer observed by CATS, but the coarser particles (division d2) appear to be distributed much lower in the atmosphere (e.g Fig 17, 19 and 20). As already mentioned, we looked at similar plots for 45 overpasses in total, and the comparison gave similar results.

In the MetUM there is a size dependence in the dust uplift scheme, where finer particles are lofted more easily. However, previous studies suggest that the MetUM division d2 dust would be expected to reach higher altitudes away from source regions than it does. The behaviour downstream from the source seems to indicate that as the dust-laden air mass moves away, the coarse particles are lost too quickly in the model prediction. This would fit with what previous studies have found, for example Ansmann et al., (2017).

**4.4. Effect of large scale wind and boundary layer height**

In this section we investigate potential drivers for the observed discrepancies in the vertical distribution of dust in the MetUM and ECMWF CAMS. This is a difficult task as there are many competing factors that influence how dust is lifted into the atmosphere and subsequently transported, and these vary considerably between models. In the MetUM the three processes which are most likely to have an impact on the vertical distribution of dust are the
convection scheme, boundary layer (BL) height at source, and the largescale wind. Looking at the largescale wind field and BL height should show whether the modelled dust layer height is controlled by the largescale wind or by boundary layer mixing processes at the source. If examination of these processes cannot explain why the dust is too low in altitude, then the most likely cause is to be researched in the convection scheme. There is, however, no direct measure of convection in the model output fields from the MetUM, and therefore any influence can only be
inferred from the data that is available to us.

Back-trajectories from HYSPLIT and NAME and SEVIRI dust RGB imagery were used to determine the central trajectory of the dust sampled during each case study from source (Fig 1). The dust concentration for each size bin, the large-scale wind (w) and the BL height were extracted from the model output along the track, and plotted as a cross-section every 6 hours from the time of uplift to the time of sampling by the aircraft.

Fig 21 displays such cross-sections for case study 3. The dust was uplifted from Mali on the 13$^{th}$ August, with a secondary uplift along the track, in Mauritania. At the time of uplift both models show a ~ 0.3 m s$^{-1}$ increase in the largescale wind velocity. An increase in largescale wind velocity at the time of uplift of between 0.2 and 0.8 m s$^{-1}$ was observed for all the cases looked at. At the time of dust uplift, the BL height was typically 4 – 5 km, and the dust mixed up to its top. The altitude to which the dust reached over the source regions of Africa compared well
with the CATS observations of the depth of the dust layer over Africa (fig 17–20). This suggests that problems with the BL height in the MetUM may not be the cause for the dust layer being represented too low in the atmosphere away from the source region.

From the data presented here it is not possible to determine how well the models represent largescale wind in the dust source regions. Previous studies which have looked at this issue more comprehensively do however suggest that there is an underprediction of wind fields in the models, which is also linked to coarse resolution modelling (eg. Chouza et al., 2016). Evan et al, (2016) showed that desert dust emission is to first order a function of wind speed, and it is against this quantity that models parametrise the dust source. This combined with our observations of an increase in largescale wind velocity at the time of dust uplift suggest that further investigation into the role of wind speed in the models would be helpful as a key part of getting the amount of dust uplift right.

## 5 Conclusions

The vertical distribution, particle size distribution, and mass concentration are the key properties that are predicted in a dust transport model. On the other hand, the main observable quantity on a global scale is aerosol optical depth, from AERONET (Holben et al., 1998), MODIS (Hsu et al, 2004, 2006, 2013; Levy et al, 2013; Sayer et al, 2013, 2014), and potentially other sources such as the Polar Multi-Sensor Aerosol product (PMAp; Lang et al, 2017), the Visible Infrared Imaging Radiometer Suite (VIIRS; Hsu et al, 2019), and several others. Aerosol optical depth is at the same time an optical property and a vertically integrated quantity, meaning that a same observable AOD can be retrieved e.g. with differing combinations of concentration and particle size distribution, or with a differing vertical distribution. It is good practice to pull the model towards the observations, and this can be achieved through tuning and through data assimilation: this means that we can expect a good model to yield a sensible prediction of the AOD. This is however insufficient to state that the underlying microphysical properties, from which AOD is derived, are correctly balanced.

The vertical distribution and particle size distribution heavily affect how dust is transported and how quickly it is deposited. Wind speed and direction are altitude dependent, meaning that transport is heavily dependent on the altitude of a layer. Residence time and transport range are affected by both the particle size distribution (coarse particles tend to be deposited more quickly) and vertical distribution (turbulent mixing in the boundary layer speeds up deposition, compared to the free troposphere). The representation of these properties in a model can affect the predicted AOD gradient across the Atlantic, for example. All this means that in the case of a model constrained by AOD observations only, other processes may need to compensate for a potential imbalance in the microphysical representation, such as e.g. the intensity of sources and sinks. The microphysical properties and the three-dimensional spatial distribution of dust are thus deeply interconnected.

We have used a combination of remote sensing and in-situ measurements to characterize the vertical distribution and transport of Saharan dust over the Eastern Atlantic and West Africa during August 2015, and to evaluate the dust forecasts from two operational global atmospheric models (MetUM and ECMWF CAMS). The dust AOD predictions at short forecast lead times from both models were in agreement with the aircraft, satellite, and AERONET observations, but with a low bias (note that both models assimilate

AOD). Previous studies resulted in similar findings, e.g. Roberts et al (2018) found that the AOD over the Sahara is well-represented, compared to MODIS, on a seasonal to monthly timescale. On the other hand, we found that the vertical distribution of aerosol extinction coefficient and dust concentration could benefit from improvements. Our results show that the predicted vertical distribution places the dust low in the atmosphere, when compared to observations. Agreement between measured and modelled profiles was better near source, with differences increasing downstream, confirming the findings of previous studies (e.g Kim et al., 2014, Ansmann et al., 2017). Similarly, Konsta et al (2018) concluded that the BSC-DREAM8b regional dust model overestimated dust extinction in the Saharan source regions, and underestimated transported dust over Europe and the Atlantic.

This issue was particularly noticeable in the MetUM, where the coarser dust was not transported high enough in the atmosphere, or far enough away from source, compared with the observations. This suggests that the model could be settling the coarse mode dust too quickly, and similar findings have also been observed in previous studies (e.g Kim et al., 2014; Mona et al., 2014; Binietoglou et al., 2015). We also found that both models underpredict the coarse mode, and overpredict the fine mode. The discrepancy between the magnitude of the measured and modelled extinction coefficient is much less than for the concentration profiles. This is likely to be due to the microphysical representation, since small particles are more optically efficient. Due to MODIS AOD data assimilation, and model tuning against AERONET observations, the large under prediction of coarse mode dust in the models is compensated with a relatively small effect on the forecast average extinction coefficient and aerosol optical depth, even with the discrepancies in size distribution and dust concentration. Our findings support a recent study by Adebiyi and Kok (2020), who reported a large underprediction of coarse mode dust in six climate models and that, for this reason, the global dust burden was underpredicted by a factor of 4. Huneeus et al (2011) also found that models tend to simulate the climatology of vertically integrated parameters (AOD and AE) much better than total deposition and surface concentration. Hoshyaripour et al (2019) also highlighted discrepancies between ICON-ART dust predictions and Multiangle Imaging Spectroradiometer (MISR) observations, associated with uncertainties in particle size distribution and emission mechanisms.

The overestimation of dust concentration in the finer ECMWF CAMS bins, and the underestimate of coarser dust is something that ECMWF are aiming to address in the future. In order to do this an updated dust emission scheme based on Remy et al (2019) using the Kok et al., (2012) estimates of size distribution at emission would be used. It is expected that this would increase the total dust concentration and shift it to the larger sizes, thus keeping total extinction similar to its present values, but more accurately representing the dust size distribution. After these changes have been implemented, a further study like the present one can help quantify the improvement introduced.

We have also investigated the processes driving dust uplift in the models, and our analysis suggests that uncertainties in the large-scale wind and the emitted size distribution are likely causes of differences between observations of the Saharan Air Layer (SAL) and MetUM predictions. The crude representation of the dust size-distribution in the MetUM 2-bin dust scheme is another important factor. The MetUM operational dust

forecast is intended to be used primarily for AOD forecasts and extinction for visibility purposes, and although improvements of the microphysical properties would be desirable, the current implementation is satisfactory to an extent, and has the advantage of being computationally cheap. We also note that the dust scheme used in the Met Office climate model differs, using 6 size bins rather than 2, with the 6-bin version yet to be evaluated as in this article.

The scheme used to represent dust microphysical properties in models deserves attention as a key element to pursue accurate mineral dust predictions. Simple schemes (such as for instance the 2-bin dust size-distribution in the operational version of the MetUM) have the obvious advantage of being viable in terms of computing resources required, but on the other hand there is the consequence of giving a less accurate representation of the microphysical properties. This could be addressed by increasing the number of variables used to represent the size distribution, for example by using a scheme with 2 or more modes, each defined by 2 variables, such as in the GLOMAP-mode aerosol scheme in UKCA (Mulcahy et al, 2020), although the ability of this scheme to represent the coarse and giant modes correctly still needs to be proven. Whatever approach is chosen, it needs to allow for coarse and giant particles to be represented, a capability currently missing in many models (Huneeus et al, 2011). It is to be noted that there are plans in place to move to GLOMAP dust within the operational Global MetUM in the near future, and also ongoing experimentation with this scheme in the ECMWF IFS within CAMS. Moreover, there are plans to modify the latter scheme by adding a third (super coarse) mode: these are changes in the right direction.

As the size-distribution affects gravitational settling, it indirectly affects the three-dimensional distribution. Additionally, some processes may deserve better attention, as studies suggest that they could increase the lifetime of coarse and giant particles beyond what is predicted for gravitational settling: e.g. turbulence within the Saharan Air Layer, particle electrification, and the role of convective systems (Van Der Does et al, 2018). The optimum balance between these processes is still to be understood, as is the correct estimation of emission intensity. The dust observable properties, in terms of the aerosol optical depth, the particle sizes, the spatial distribution, and the vertical distribution, are determined by these processes. The combination of all these properties determines the impact of dust on the climate system, hence the importance of understanding these processes better (see e.g. Kok et al, 2017).

Two more points that need attention are the particle shape and effect of dust on the radiation field, atmospheric heating rates and thermodynamics and the dust transport itself. If dust particles are assumed spherical in the dust transport models, many computations are easier, however it is well-known that dust particles are very irregular. The mass-to-extinction conversion and the drag coefficient calculations (which affect deposition and transport) are directly affected by particle shape. Moreover. dust microphysics and consequent radiative properties such as single-scattering albedo and asymmetry parameter do alter the computations of atmospheric radiation due to dust. In turn, this affects the heating rates of atmospheric layers, the atmospheric thermodynamics, the convective motions, and the wind fields, which result in possible modifications of the dust transport patterns. An improvement of the radiative transfer models within dust models is therefore suggested, to integrate the latest understanding of dust microphysics.

As this study highlights the limitations ascribed to using AOD as the main observable quantity towards which to verify, tune and pull the model, it also supports the perspective of improving the set of aerosol observations that can be used on a global scale. In particular, observational datasets exist for the vertical dust distribution, which can be exploited to better constrain the predictions. The most obvious one is the CALIPSO dataset, which has been observing the global aerosol distribution since 2006 (Winker et al, 2010; Liu et al, 2008; Tsamalis et al, 2013), and in the future EarthCARE is expected to be another very good candidate (Illingworth et al, 2015). Note that this perspective is not limited to using active sensors, and studies exist on the observation of the vertically resolved distribution from passive hyperspectral instruments in the infrared (Callewaert et al, 2019). In the long-term, providing observations not only of AOD, but also on the vertical distribution of aerosols, could become the driver for operational space missions.

In addition to vertically-resolved information, we also highlight the importance and need for better constrained size-resolved properties of dust, needed to reproduce the correct relationship between concentration and extinction coefficient. Particle size-distributions, both in the model representation and in the observations, should cover the whole size spectrum, including the giant mode (Marenco et al, 2018; Ryder et al, 2019). Ideally, these observations should be coordinated, vertically resolved, and established across a number of locations downstream from sources, e.g. across the Tropical Atlantic. Sporadic observations do exist, we are here advocating a more systematic approach. For instance, a number of balloonborne sensors are being developed and could be used for this purpose (see e.g. Renard et al, 2016; Fujiwara et al, 2016; Smith et al, 2019; Dagsson-Waldhauserova, 2019).

To conclude, we highlight how campaigns focusing on a combination of in-situ and remote sensing observations can provide information to at the same time validate existing model developments and help identify the areas requiring developments. In the last few years, considerable improvements have been made to operational dust forecasts, and with this paper we want to contribute to this effort by (1) indicating a few points that could be addressed in the models, and (2) provide a few datasets and a selection of case studies for future model assessments.

## 6 Acknowledgements

Airborne data were obtained using the BAe-146-301 Atmospheric Research Aircraft operated by Directflight Ltd and managed by the Facility for Airborne Atmospheric Measurements (FAAM).

The staff of the Met Office, the Universities of Leeds, Manchester and Hertfordshire, FAAM, Direct Flight, Avalon Engineering and BAE Systems are thanked for their dedication in making the ICE-D and AER-D campaigns a success. Claire Ryder acknowledges NERC support through Independent Research Fellowship NE/M018288/1. The authors thank the principal investigators and their staff for establishing and maintaining the AERONET sites used in this study. The MODIS data in this study were acquired as part of NASA's Earth Science Enterprise. The algorithms were developed by the MODIS Science Teams and the data were processed by the MODIS Adaptive Processing System (MODAPS) and Goddard Distributed Archive Centre (DAAC) and are archived and distributed by the Goddard DAAC. The authors gratefully acknowledge the NOAA Air Resources Laboratory for the provision of the HYSPLIT transport and dispersion model used in this publication.

The FAAM aircraft datasets collected during the ICE-D and AER-D campaigns are available from the British Atmospheric Data Centre, Centre for Environmental Data Analysis, at the following URL: http://catalogue.ceda.ac.uk/uuid/d7e02c75191a4515a28a208c8a069e70 (Bennett, 2019).

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

# Tables

| Flight | Flight section | Time | Lat N | Lon W | Aerosol extinction coefficient (Mm$^{-1}$) | | | |
|---|---|---|---|---|---|---|---|---|
| | | | | | Data source | Mean | Stdev | Max |
| B920 7 Aug | R1 | 14:29:28 to 15:00:25 | 15.56 to 17.54 | 22.98 to 21.40 | ECMWF MetUM Lidar | 55 58 57 | 38 41 47 | 126 177 329 |
| | R6 | 17:29:11 to 18:00:18 | 15.54 to 17.54 | 22.99 to 21.40 | ECMWF MetUM Lidar | 55 58 56 | 38 41 40 | 126 177 212 |
| B923 12 Aug | R1 | 09:18:08 to 11:51:28 | 16.03 to 27.30 | 22.97 to 13.82 | ECMWF MetUM Lidar | 140 90 110 | 120 120 130 | 490 720 1130 |
| B924 12 Aug | R3-R4 | 15:11:57 to 16:04:43 | 23.26 to 24.49 | 18.81 to 17.81 | ECMWF MetUM Lidar | 169 51 180 | 97 27 180 | 485 205 1260 |
| | R6-R7 | 16:48:29 to 18:33:01 | 16.44 to 24.08 | 23.03 to 18.18 | ECMWF MetUM Lidar | 107 46 60 | 84 35 100 | 443 169 1150 |
| B927 15 Aug | R1 | 13:59:06 to 14:46:00 | 11.42 to 15.05 | 24.55 to 23.37 | ECMWF MetUM Lidar | 81 54 78 | 60 41 96 | 332 159 372 |
| B932 20 Aug | R1 | 09:52:41 to 10:35:23 | 17.72 to 20.67 | 21.19 to 18.93 | ECMWF MetUM Lidar | 140 140 76 | 120 130 81 | 500 620 395 |

Table 1: Summary of the high-level sections from each of the flights used here. Flight sections are labelled with the letter R (runs), see text. All times UTC.

| Flight | Flight section | Time | Lat N | Lon W | Altitude AMSL (km) |
|---|---|---|---|---|---|
| B920 7 Aug | P1 | 14:03:33 to 14:25:21 | 14.94 to 15.76 | 22.78 to 23.48 | 0.1 to 6.5 |
| | P2 | 15:02:59 to 15:24:05 | 16.26 to 17.43 | 21.37 to 22.38 | 0.1 to 6.5 |
| | P7 | 17:08:08 to 17:27:50 | 17.34 to 17.94 | 21.00 to 21.53 | 0.1 to 6.5 |
| B923 12 Aug | P1 | 11:51:28 to 12:09:28 | 27.30 to 28.44 | 13.71 to 13.87 | 0.1 to 6.9 |
| B932 20 Aug | P4 | 10:37:23 to 11:01:22 | 20.01 to 20.30 | 18.88 to 20.22 | 0.1 to 6.5 |

Table 2: Summary of the aircraft profiles from each of the flights used here. Flight sections are labelled with the
letter P (profiles), see text. All times UTC.

**Figure captions**

Figure 1. Flight tracks and dust source locations (circles). Dotted lines between flight tracks and circles show the approximate mean trajectory based on SEVIRI RGB dust images, NAME and HYSPLIT back trajectories. Note that flights B923/B924 sampled the same dust event.

Figure 2: Flowchart of comparison methodology.

Figure 3: Case study 1, B920, 7$^{th}$ August, R6: (a-d) Vertical cross –section along the flight track showing the aerosol extinction coefficient for the CATS lidar (a), ECMWF CAMS (b), MetUM (c) and the aircraft lidar (d),
the colour scale is the same for all four plots. (e) ECMWF AOD map, (f) MetUM AOD map and (g) AOD map from combined observations from MODIS, AERONET (stars) and aircraft lidar (dots).

Figure 4: Case Study 1, B920, 7$^{th}$ August, R6; (a) Mean and standard deviation of the airborne lidar (green),
MetUM (red) and ECMWF (blue) extinction profiles. (b) Modelled MetUM dust concentration for divisions 1 (dark red) and 2 (red), and modelled ECMWF concentration for divisions 1 (dark blue), 2 (blue) and 3 (light blue) dust concentration. See text for the description of the divisions.


Figure 5: Case Study 1, B920, 7th August, P1. (a) Dust concentration measured by the in-situ instruments on the aircraft for MetUM dust divisions 1 (red) and 2 (green), and the total dust concentration measured (black). The division 1 and 2 concentration from the model is shown in a lighter shade of red and green respectively, with markers and error bars showing the standard deviation. (b) The right hand plot shows the same thing but for the ECMWF CAMS size bins, with the measurements shown using lines, and the model values with lines and markers for divisions 1 (red), 2 (green), and 3 (blue). See text for the description of the divisions.

Figure 6: Case Study 1, B920, 7th August, P2. (a) Dust concentration measured by the in-situ instruments on the aircraft for MetUM dust divisions 1 (red) and 2 (green), and the total dust concentration measured (black). The division 1 and 2 concentration from the model is shown in a lighter shade of red and green respectively, with markers and error bars showing the standard deviation. (b) The right hand plot shows the same thing but for the ECMWF CAMS size bins, with the measurements shown using lines, and the model values with lines and markers for divisions 1  (red), 2 (green), and 3 (blue).

Figure 7: Case Study 1, B920, 7th August, P2. (a) water vapour mixing ratio from the aircraft measurements in the profile (green), compared with the MetUM (red) and ECMWF (blue). (b) the same but for temperature – here there are two measurements of temperature shown which are in good agreement.

Figure 8: Case Study 1, B920, 7th August, P7. (a) water vapour mixing ratio from the aircraft measurements in the profile (green), compared with the MetUM (red) and ECMWF CAMS (blue). (b) the same but for temperature – here there are two measurements of temperature shown which are in good agreement.

Figure 9: Case Study 2: B923, 12th August, R1: (a-c) Vertical cross –section along the flight track showing the aerosol extinction coefficient for ECMWF CAMS (a), MetUM (b) and the aircraft lidar (c), the colour scale is the same for all three plots. (d) ECMWF CAMS AOD map, (e) MetUM AOD map and (f) AOD map from combined observations from MODIS, AERONET (stars) and aircraft lidar (dots).

Figure 10: Case Study 2: B923, 12th August, R1: (a) Mean and standard deviation of the lidar (green), MetUM (red) and ECMWF (blue) extinction profiles. (b) Modelled MetUM dust concentration for divisions 1 (dark red) and 2 (red) and modelled ECMWF concentration for divisions 1 (dark blue), 2 (blue) and 3 (light blue) dust concentration.

Figure 11: Case Study 2: B923, 12th August, P1 (landing in Fuerteventura). (a) Dust concentration measured by the in-situ instruments on the aircraft for two MetUM dust divisions 1 (red), and 2 (green), and the total dust concentration measured (black). The division 1 and 2 concentration from the model is shown in a lighter shade of red and green respectively, with markers and error bars showing the standard deviation. (b) The right-hand plot shows the same thing but for the ECMWF CAMS size bins, with the measurements shown using lines, and the model values with lines and markers for divisions 1 (red), 2 (green), and 3 (blue).

Figure 12: Case Study 3: B927, 15th August, R1: (a-c) Vertical cross –section along the flight track showing the aerosol extinction coefficient for ECMWF CAMS (a), MetUM (b) and the aircraft lidar (c), the colour scale is the same for these three plots. (d) ECMWF AOD map, (e) MetUM AOD map and (f) AOD map from combined observations from MODIS, AERONET (stars) and aircraft lidar (dots).

Figure 13: Case Study 3: B927, 15th August, R1: (a) Mean and standard deviation of the lidar (green), MetUM (red) and ECMWF (blue) extinction profiles. (b) Modelled MetUM dust concentration for divisions 1 (dark red) and 2 (red), and modelled ECMWF concentration for divisions 1 (dark blue), 2 (blue) and 3 (light blue) dust concentration.

Figure 14: Case Study 4: B932, 20th August, R1: (a-c) Vertical cross –section along the flight track showing the aerosol extinction coefficient for ECMWF CAMS (a), MetUM (b) and the aircraft lidar (c), the colour scale is the same for all four plots. (d) ECMWF AOD map, (e) MetUM AOD map and (f) AOD map from combined observations from MODIS, AERONET (stars) and aircraft lidar (dots).

Figure 15: Case Study 4: B932, 20<sup>th</sup> August, R1: (a) Mean and standard deviation of the lidar (green), MetUM (red) and ECMWF (blue) extinction profiles. (b) Modelled MetUM concentration for divisions 1 (dark red), 2 (red), and modelled ECMWF concentration for divisions 1 (dark blue), 2 (blue) and 3 (light blue) dust concentration.

Figure 16: Case Study 4: B932, 20<sup>th</sup> August, P4. (a) Dust concentration measured by the in-situ instruments on the aircraft for two MetUM dust divisions 1 (red), and 2 (green), and the total dust concentration measured (black). The division 1 and 2 concentration from the model is shown in a lighter shade of red and green respectively, with markers and error bars showing the standard deviation. (b) The right hand plot shows the same thing but for the ECMWF CAMS size bins, with the measurements shown using lines, and the model values with lines and markers for divisions 1 (red), 2, (green), and 3 (blue).

Figure 17: (a-c) CATS and (d-f) MetUM data for 00z on the 7<sup>th</sup> August, in the form of vertical cross–sections along the satellite track: (a) CATS extinction coefficient; (b) CATS feature type; (c) CATS overpass track; (d) MetUM total dust extinction coefficient; (e) MetUM d1 dust extinction coefficient; and (f) MetUM d2 dust extinction coefficient.

Figure 18: (a-c) CATS and (d-f) MetUM data for 18z on the 7th August, in the form of vertical cross–sections along the satellite track: (a) CATS extinction coefficient; (b) CATS feature type; (c) CATS overpass track; (d) MetUM total dust extinction coefficient; (e) MetUM d1 dust extinction coefficient; and (f) MetUM d2 dust extinction coefficient. Flight B920 is simultaneous to this satellite overpass near the Cape Verde islands.

1085 Figure 19: (a-c) CATS and (d-f) MetUM data for 00z on the 8th August, in the form of vertical cross–sections along the satellite track: (a) CATS aerosol extinction coefficient; (b) CATS feature type; (c) CATS overpass track; (d) MetUM total dust extinction coefficient; (e) MetUM d1 dust extinction coefficient; and (f) MetUM d2 dust extinction coefficient.

1090

Figure 20: (a-c) CATS and (d-f) MetUM data for 00z on the $10^{th}$ August, in the form of vertical cross–sections along the satellite track: (a) CATS aerosol extinction coefficient; (b) CATS feature type; (c) CATS overpass track; (d) MetUM total dust extinction coefficient; (e) MetUM d1 dust extinction coefficient; and (f) MetUM d2 dust extinction coefficient.

1095

Figure 21: Contribution to the extinction coefficient by MetUM dust divisions d1 and d2 (top two rows), MetUM Westerly wind component, and ECMWF CAMS largescale wind. These cross-sections are extracted along the dust trajectory shown in fig 1, for case study 3 (flight B927, $15^{th}$ August).

1100

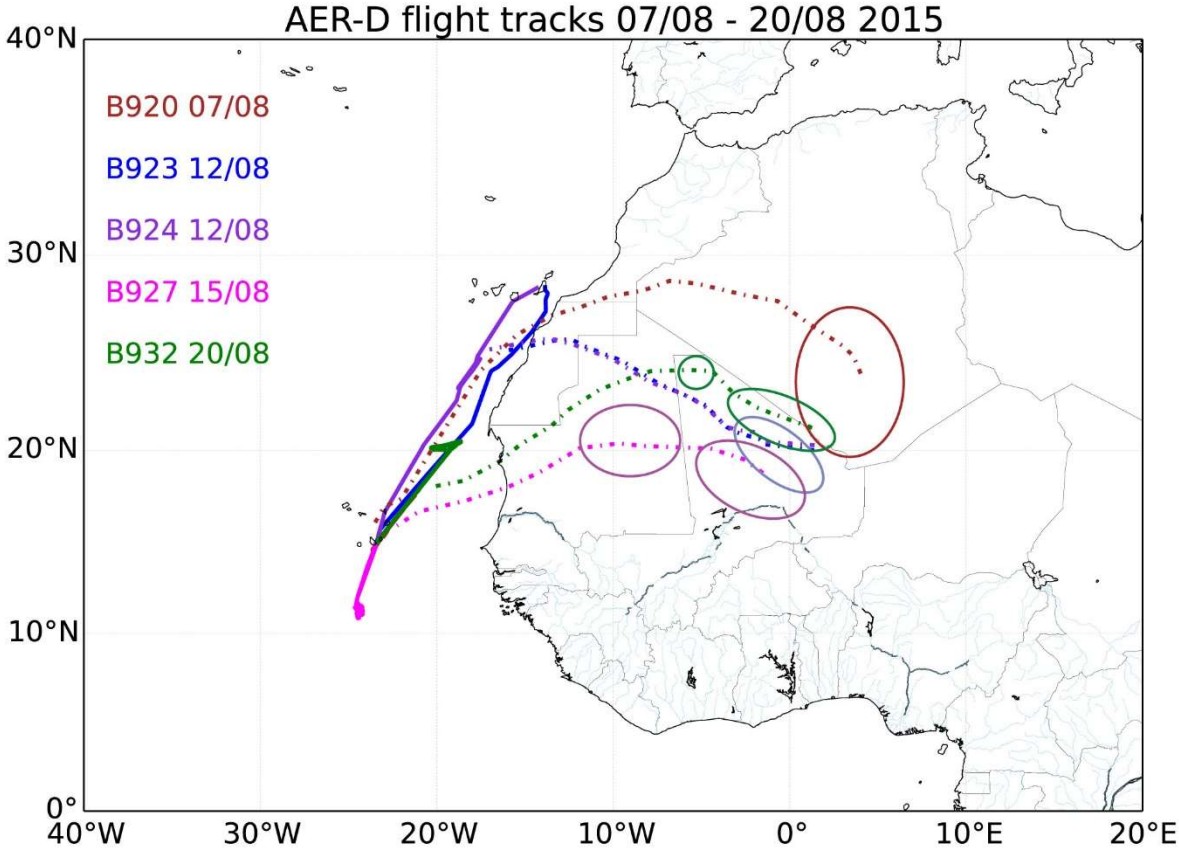

Figure 1. Flight tracks and dust source locations (circles). Dotted lines between flight tracks and circles show the approximate mean trajectory based on SEVIRI RGB dust images, NAME and HYSPLIT back trajectories. Note that flights B923/B924 sampled the same dust event.

A

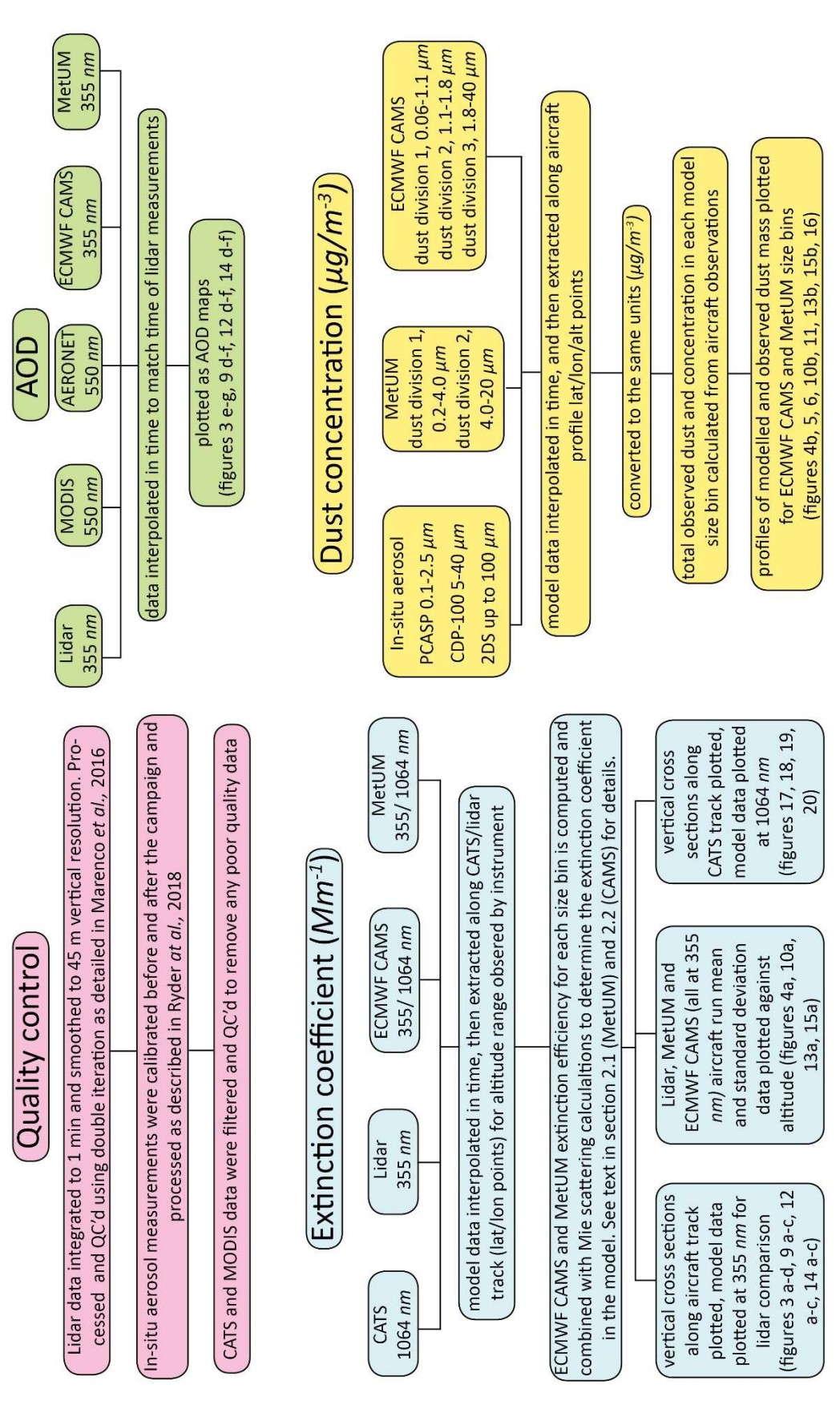

Figure 2: Flowchart of comparison methodology.

B

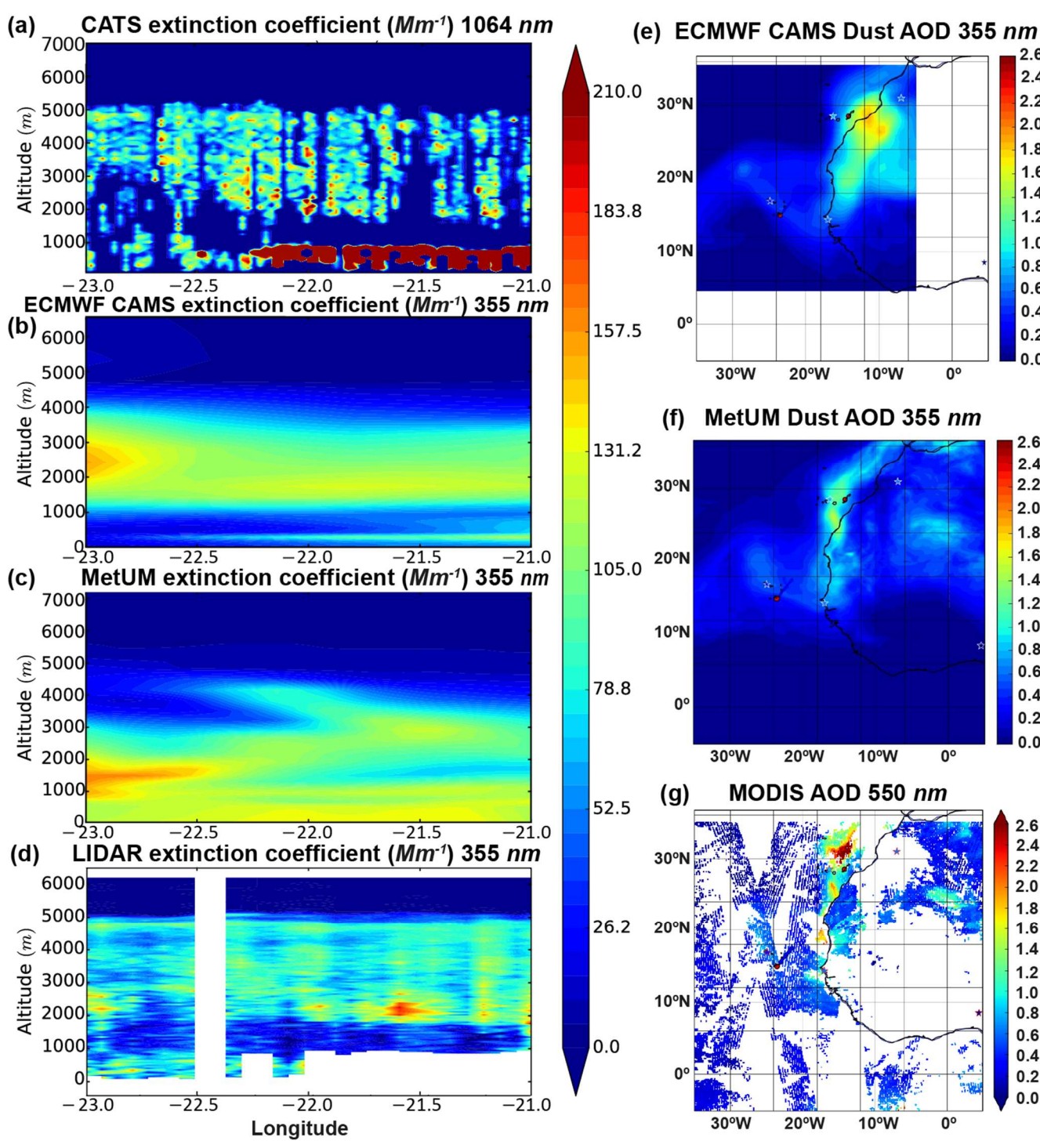

Figure 3: Case study 1, B920, 7th August, R6: (a-d) Vertical cross –section along the flight track showing the aerosol extinction coefficient for the CATS lidar (a), ECMWF CAMS (b), MetUM (c) and the aircraft lidar (d), the colour scale is the same for all four plots. (e) ECMWF AOD map, (f) MetUM AOD map and (g) AOD map from combined observations from MODIS, AERONET (stars) and aircraft lidar (dots).

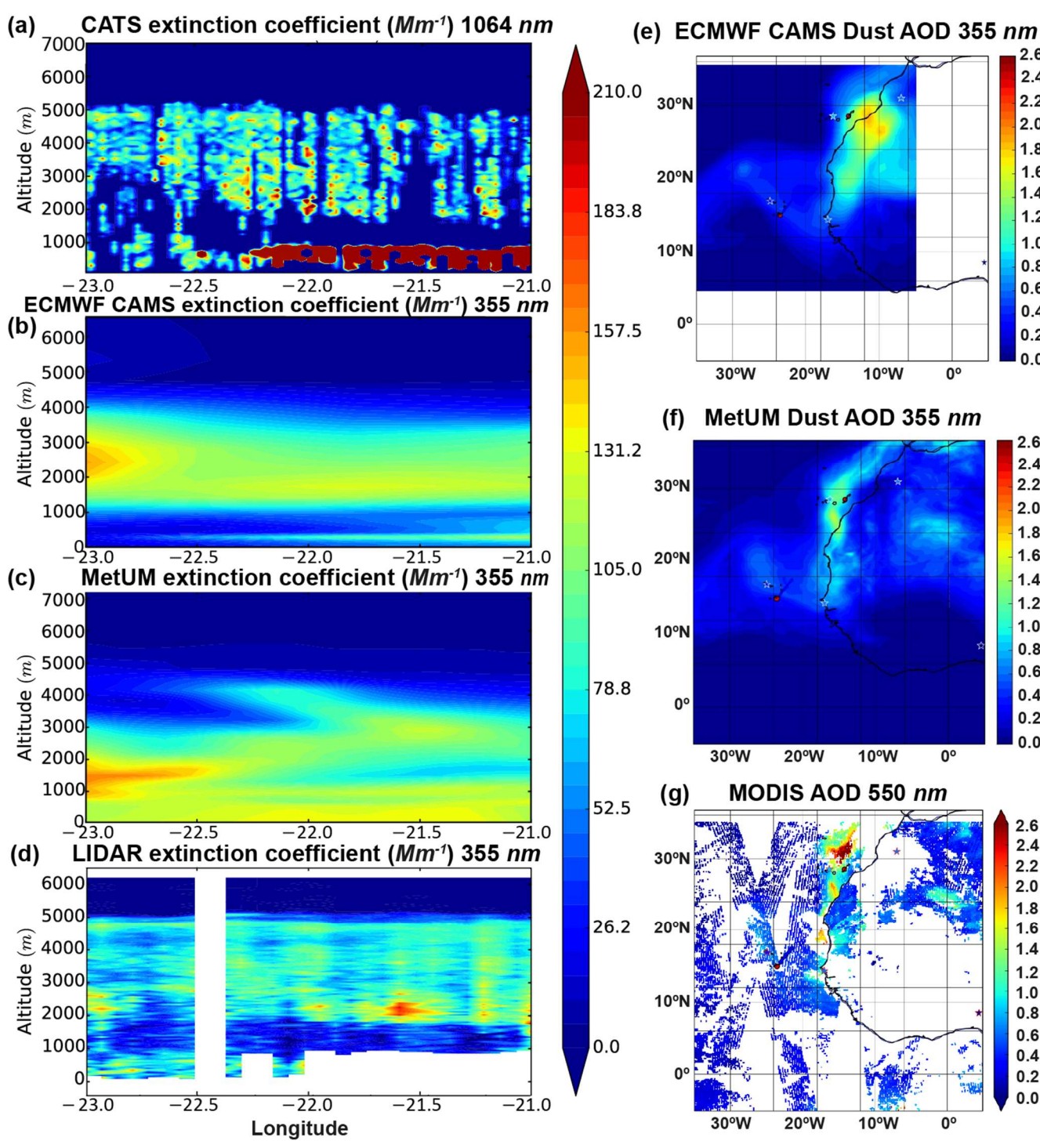

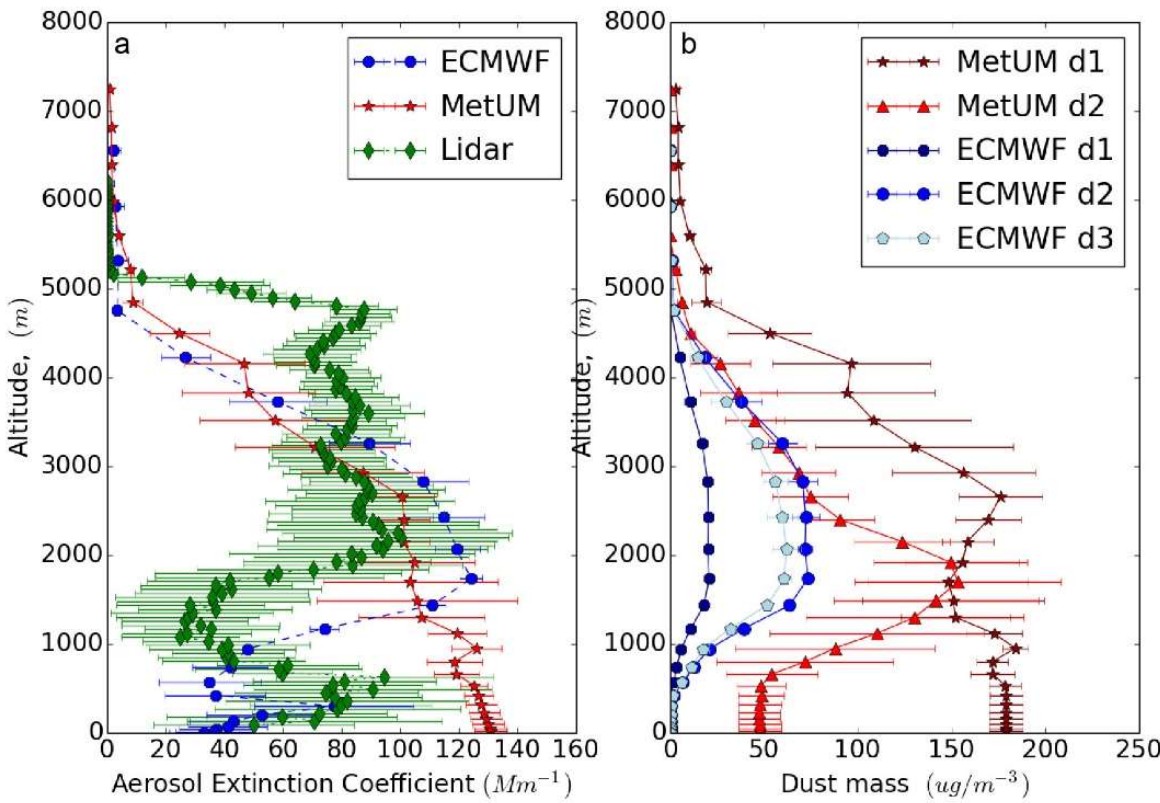

Figure 4: Case Study 1, B920, 7<sup>th</sup> August, R6; (a) Mean and standard deviation of the airborne lidar (green), MetUM (red) and ECMWF (blue) extinction profiles. (b) Modelled MetUM dust concentration for divisions 1 (dark red) and 2 (red), and modelled ECMWF concentration for divisions 1 (dark blue), 2 (blue) and 3 (light blue) dust concentration. See text for the description of the divisions.

D

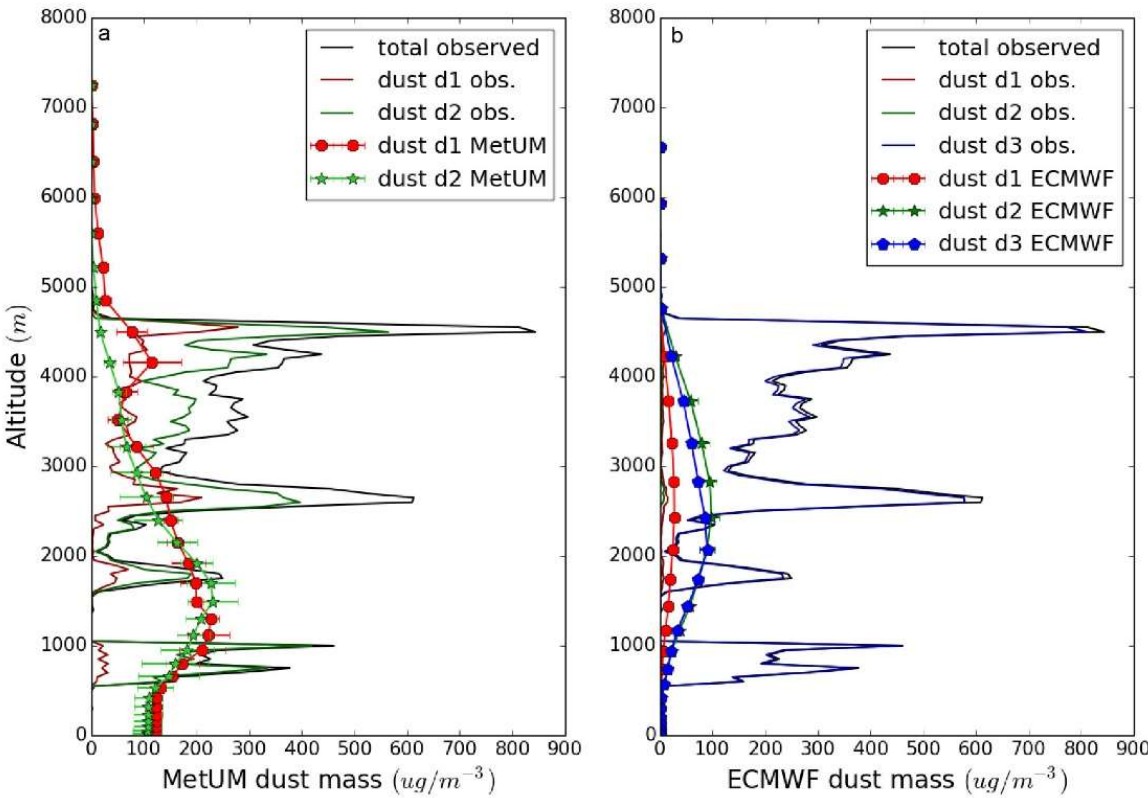

Figure 5: Case Study 1, B920, 7th August, P1. (a) Dust concentration measured by the in-situ instruments on the aircraft for MetUM dust divisions 1 (red) and 2 (green), and the total dust concentration measured (black). The division 1 and 2 concentration from the model is shown in a lighter shade of red and green respectively, with markers and error bars showing the standard deviation. (b) The right hand plot shows the same thing but for the ECMWF CAMS size bins, with the measurements shown using lines, and the model values with lines and markers for divisions 1 (red), 2 (green), and 3 (blue). See text for the description of the divisions.

E

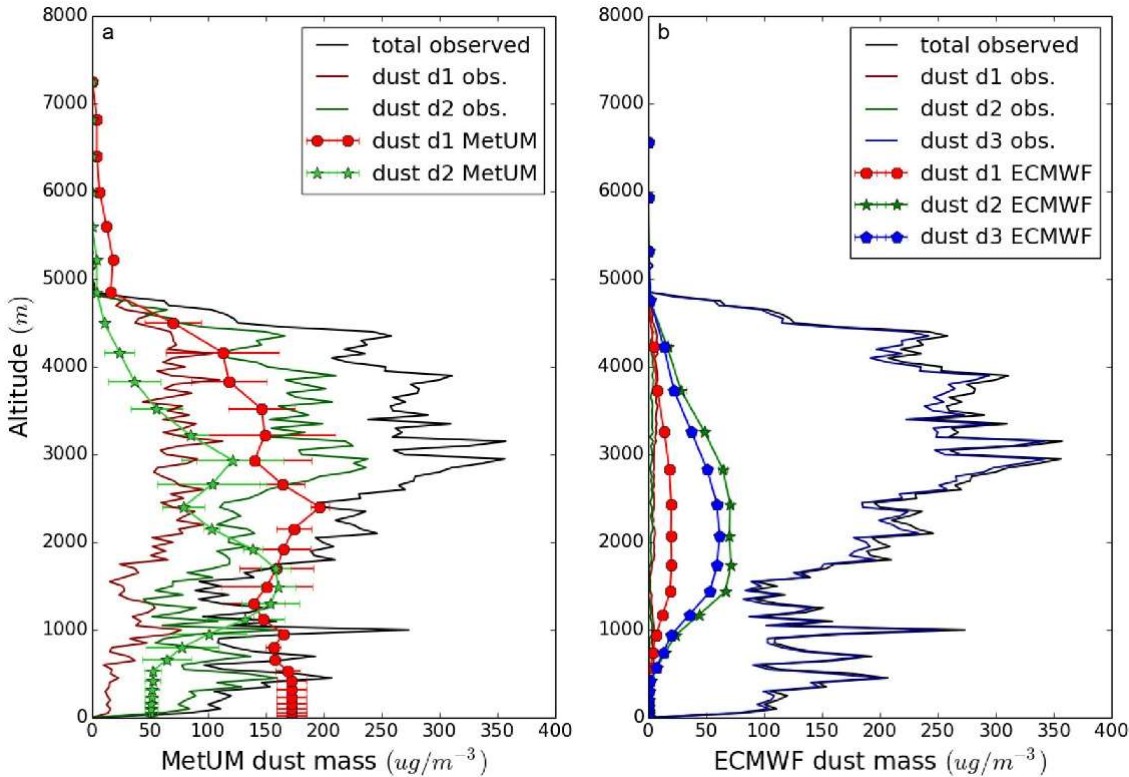

Figure 6: Case Study 1, B920, 7th August, P2. (a) Dust concentration measured by the in-situ instruments on the aircraft for MetUM dust divisions 1 (red) and 2 (green), and the total dust concentration measured (black). The division 1 and 2 concentration from the model is shown in a lighter shade of red and green respectively, with markers and error bars showing the standard deviation. (b) The right hand plot shows the same thing but for the ECMWF CAMS size bins, with the measurements shown using lines, and the model values with lines and markers for divisions 1 (red), 2 (green), and 3 (blue).

F

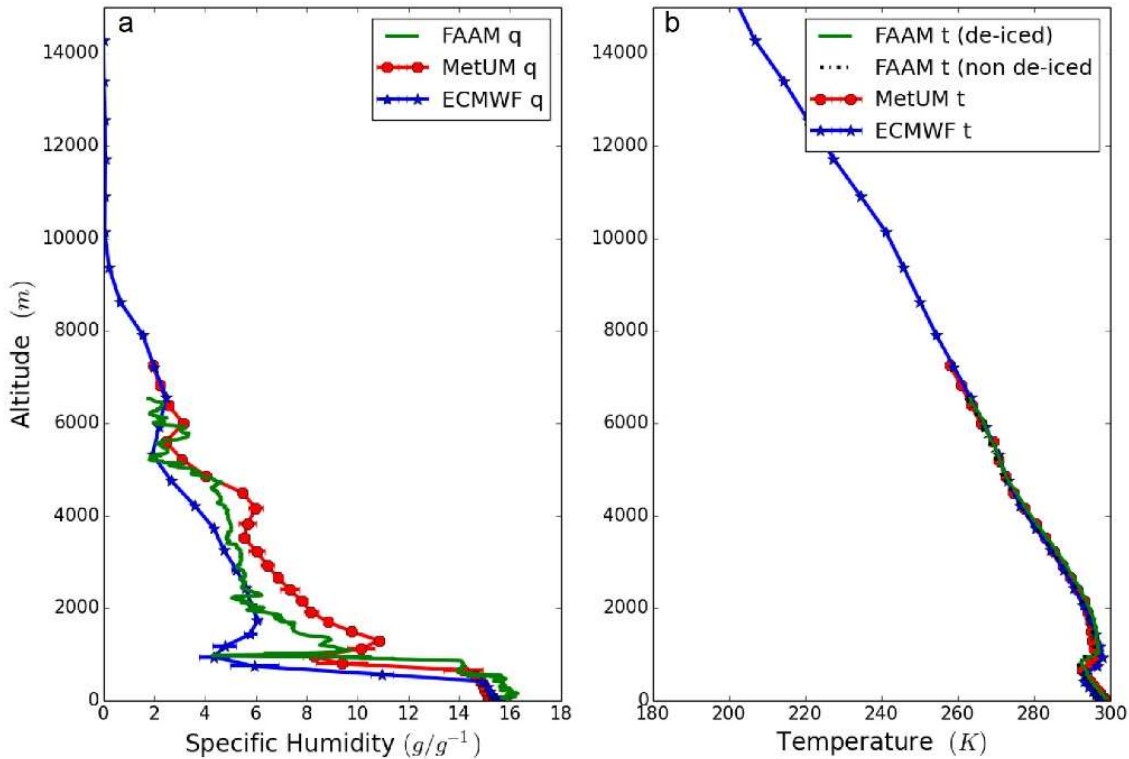

Figure 7: Case Study 1, B920, 7th August, P2. (a) water vapour mixing ratio from the aircraft measurements in the profile (green), compared with the MetUM (red) and ECMWF (blue). (b) the same but for temperature – here there are two measurements of temperature shown which are in good agreement.

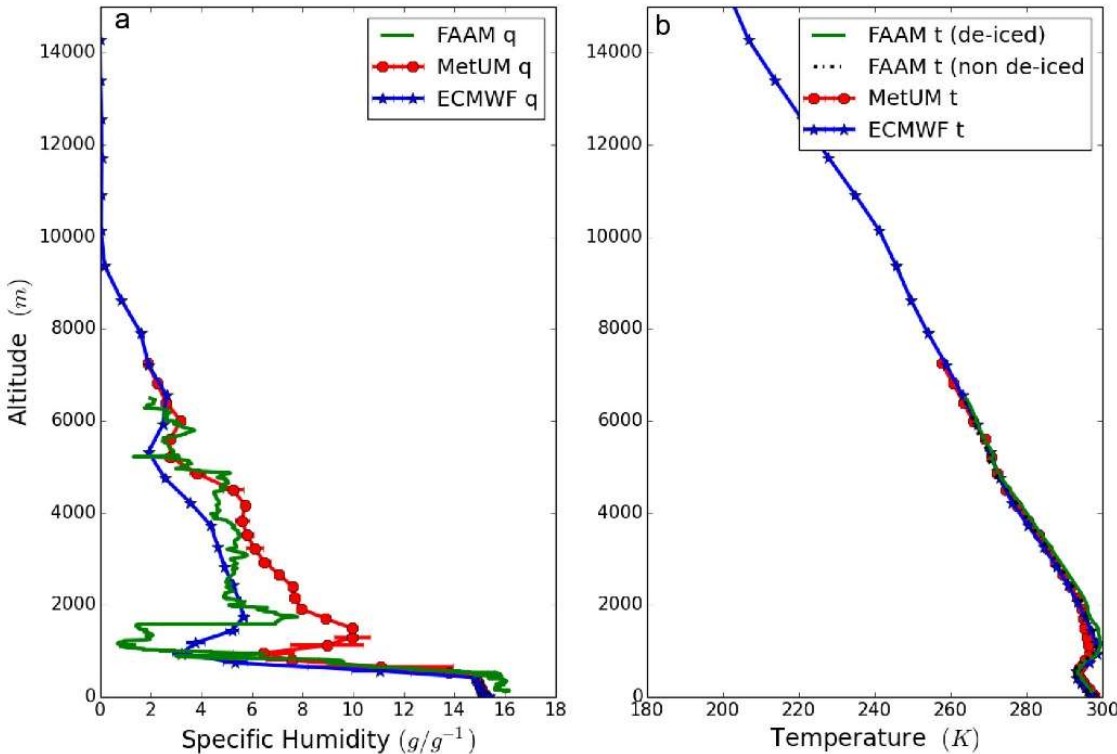

Figure 8: Case Study 1, B920, 7th August, P7. (a) water vapour mixing ratio from the aircraft measurements in the profile (green), compared with the MetUM (red) and ECMWF CAMS (blue). (b) the same but for temperature – here there are two measurements of temperature shown which are in good agreement.

H

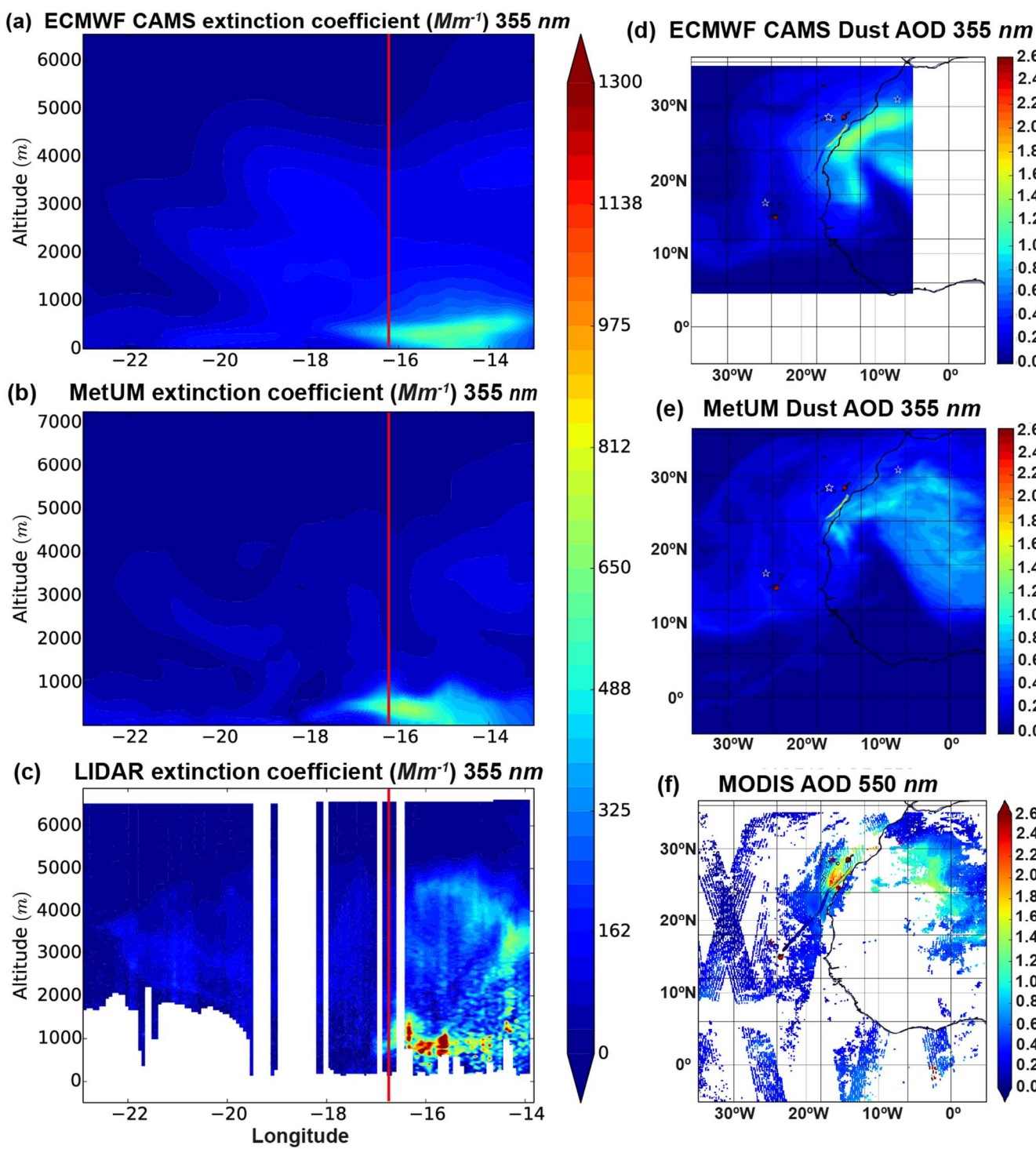

Figure 9: Case Study 2: B923, 12$^{th}$ August, R1: (a-c) Vertical cross –section along the flight track showing the aerosol extinction coefficient for ECMWF CAMS (a), MetUM (b) and the aircraft lidar (c), the colour scale is the same for all three plots. (d) ECMWF CAMS AOD map, (e) MetUM AOD map and (f) AOD map from combined observations from MODIS, AERONET (stars) and aircraft lidar (dots).

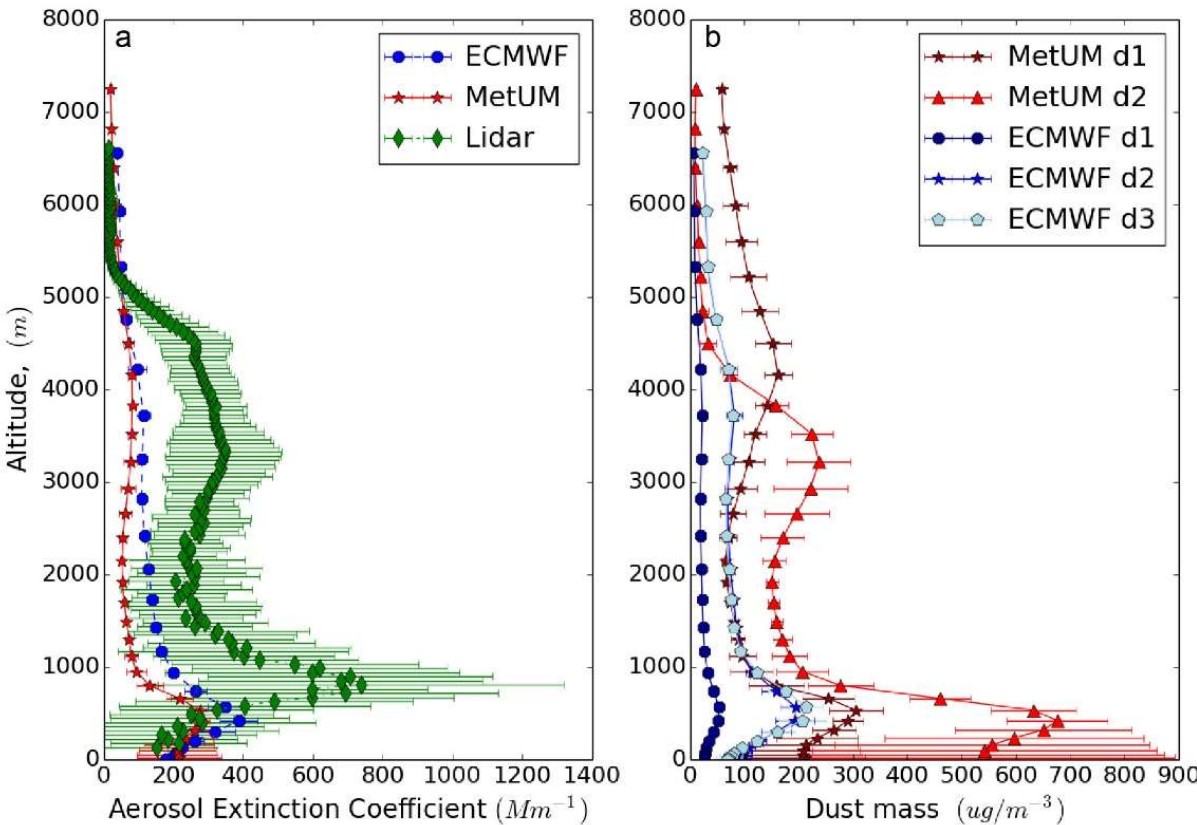

Figure 10: Case Study 2: B923, 12th August, R1: (a) Mean and standard deviation of the lidar (green), MetUM (red) and ECMWF (blue) extinction profiles. (b) Modelled MetUM dust concentration for divisions 1 (dark red) and 2 (red) and modelled ECMWF concentration for divisions 1 (dark blue), 2 (blue) and 3 (light blue) dust concentration.

J

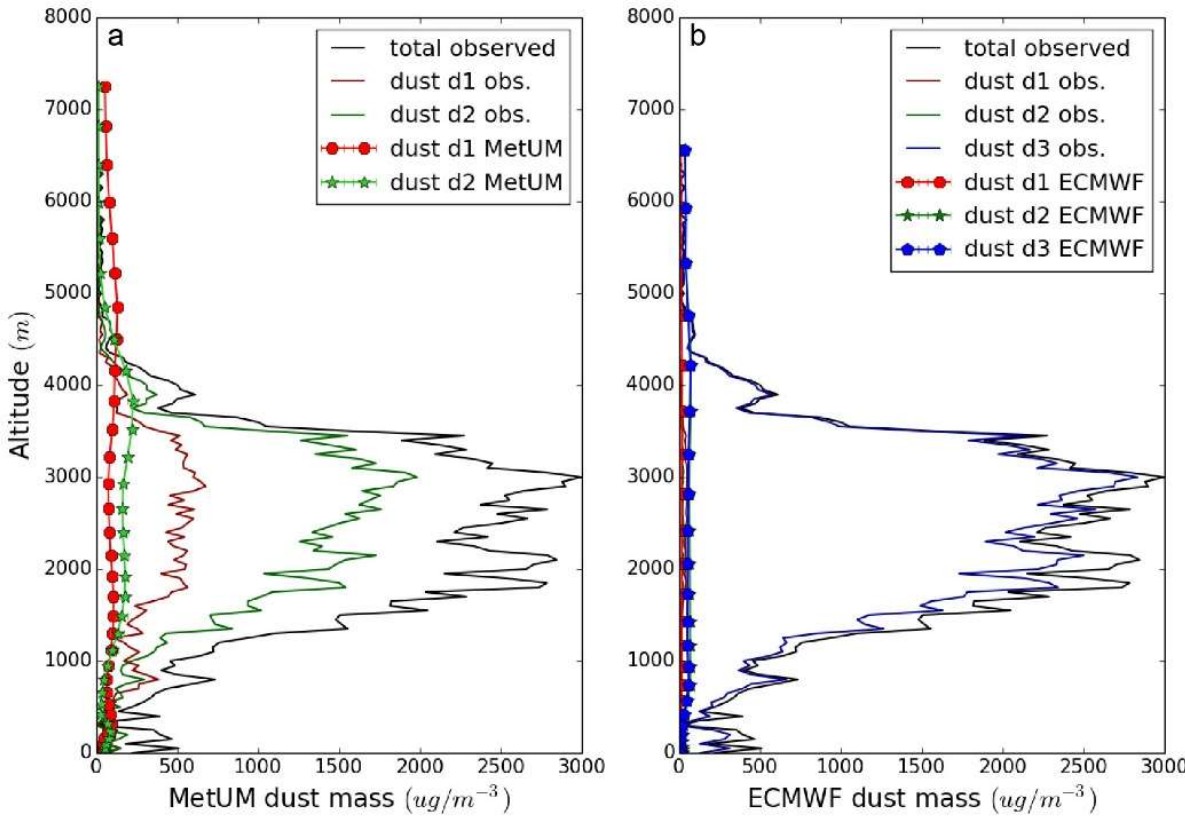

Figure 11: Case Study 2: B923, 12th August, P1 (landing in Fuerteventura). (a) Dust concentration measured by the in-situ instruments on the aircraft for two MetUM dust divisions 1 (red), and 2 (green), and the total dust concentration measured (black). The division 1 and 2 concentration from the model is shown in a lighter shade of red and green respectively, with markers and error bars showing the standard deviation. (b) The right-hand plot shows the same thing but for the ECMWF CAMS size bins, with the measurements shown using lines, and the model values with lines and markers for divisions 1 (red), 2 (green), and 3 (blue).

K

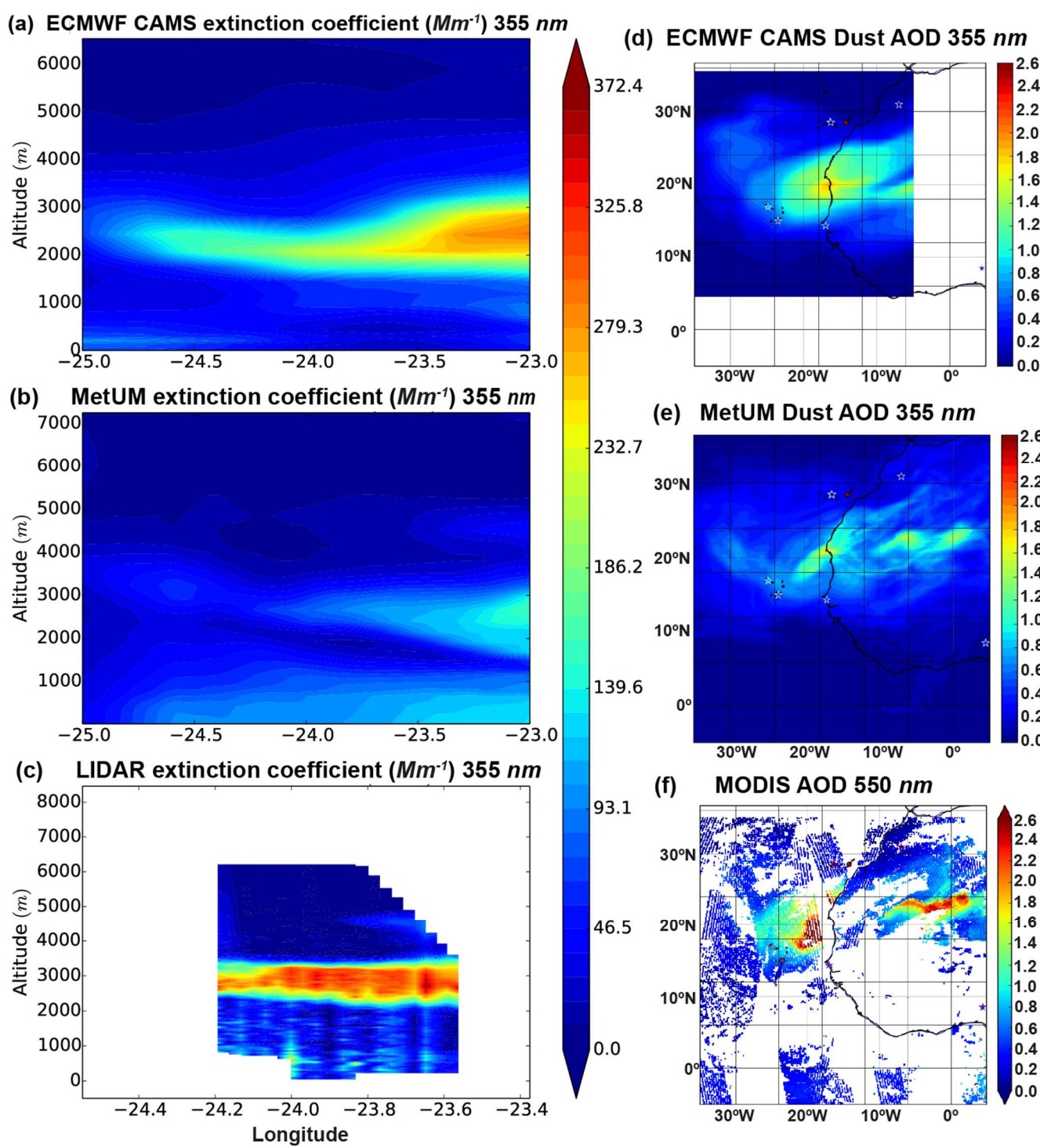

Figure 12: Case Study 3: B927, 15[th] August, R1: (a-c) Vertical cross –section along the flight track showing the aerosol extinction coefficient for ECMWF CAMS (a), MetUM (b) and the aircraft lidar (c), the colour scale is the same for these three plots. (d) ECMWF AOD map, (e) MetUM AOD map and (f) AOD map from combined observations from MODIS, AERONET (stars) and aircraft lidar (dots).

L

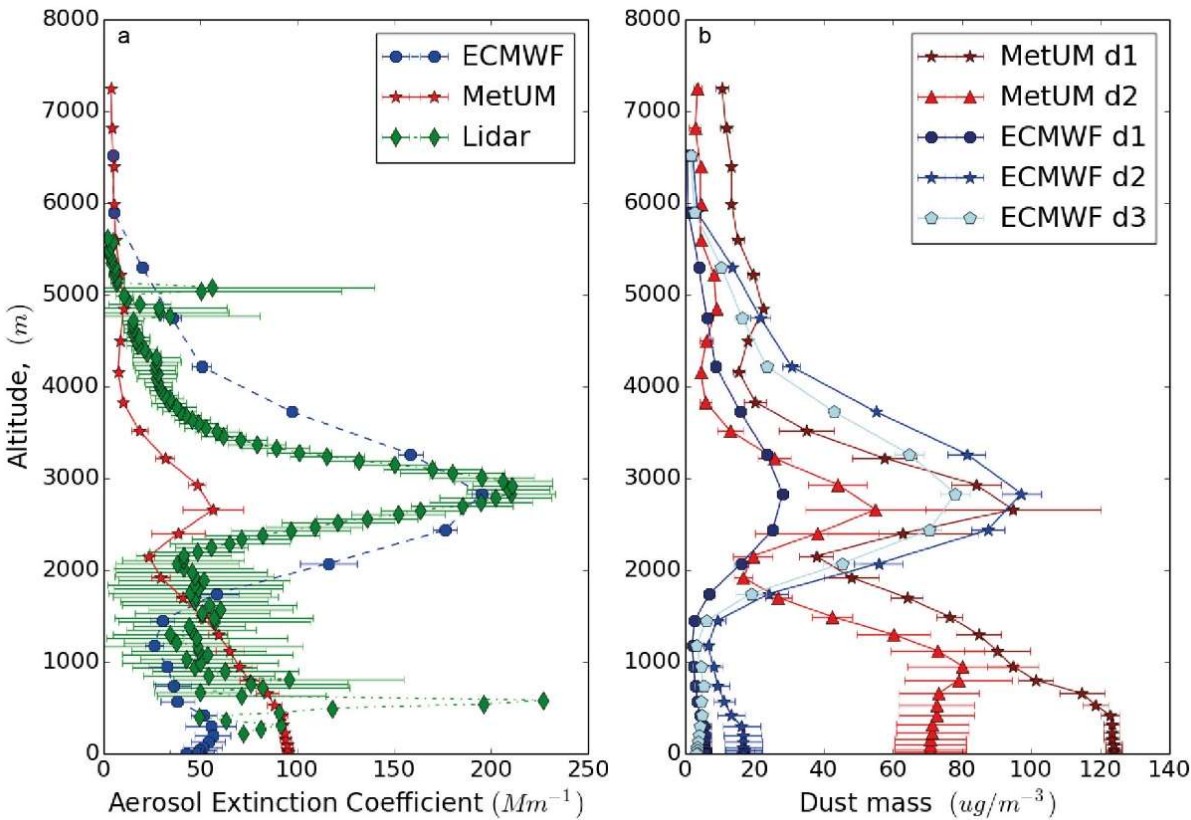

Figure 13: Case Study 3: B927, 15th August, R1: (a) Mean and standard deviation of the lidar (green), MetUM (red) and ECMWF (blue) extinction profiles. (b) Modelled MetUM dust concentration for divisions 1 (dark red) and 2 (red), and modelled ECMWF concentration for divisions 1 (dark blue), 2 (blue) and 3 (light blue) dust concentration.

M

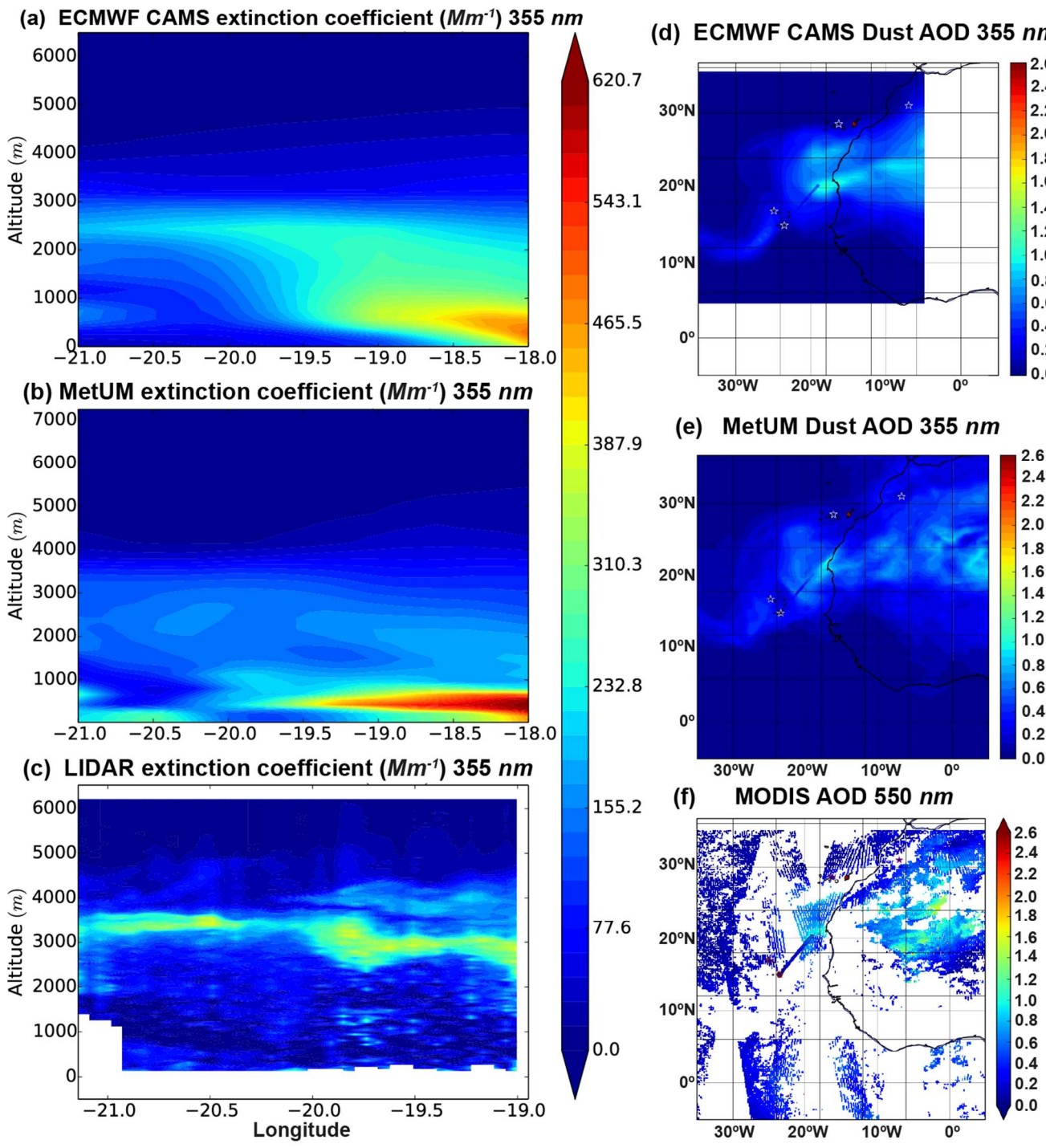

Figure 14: Case Study 4: B932, 20th August, R1: (a-c) Vertical cross –section along the flight track showing the aerosol extinction coefficient for ECMWF CAMS (a), MetUM (b) and the aircraft lidar (c), the colour scale is the same for all four plots. (d) ECMWF AOD map, (e) MetUM AOD map and (f) AOD map from combined observations from MODIS, AERONET (stars) and aircraft lidar (dots).

N

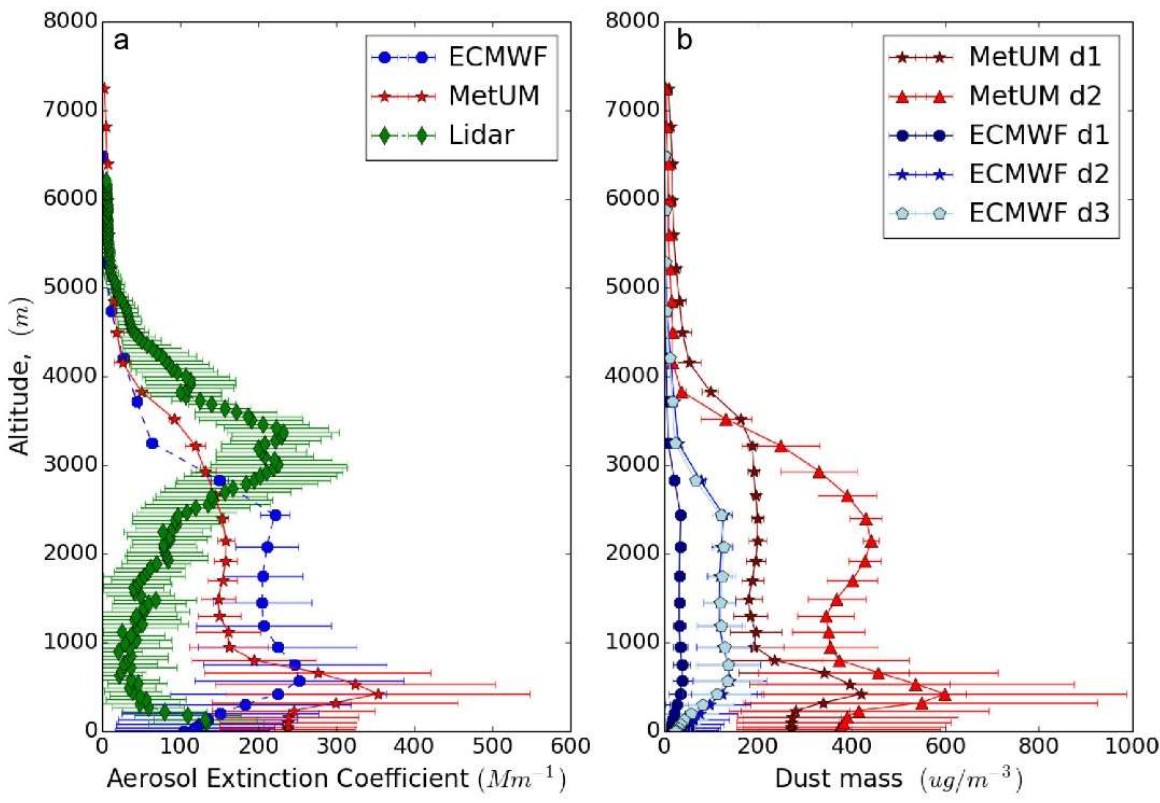

Figure 15: Case Study 4: B932, 20th August, R1: (a) Mean and standard deviation of the lidar (green), MetUM (red) and ECMWF (blue) extinction profiles. (b) Modelled MetUM concentration for divisions 1 (dark red), 2 (red), and modelled ECMWF concentration for divisions 1 (dark blue), 2 (blue) and 3 (light blue) dust concentration.

O

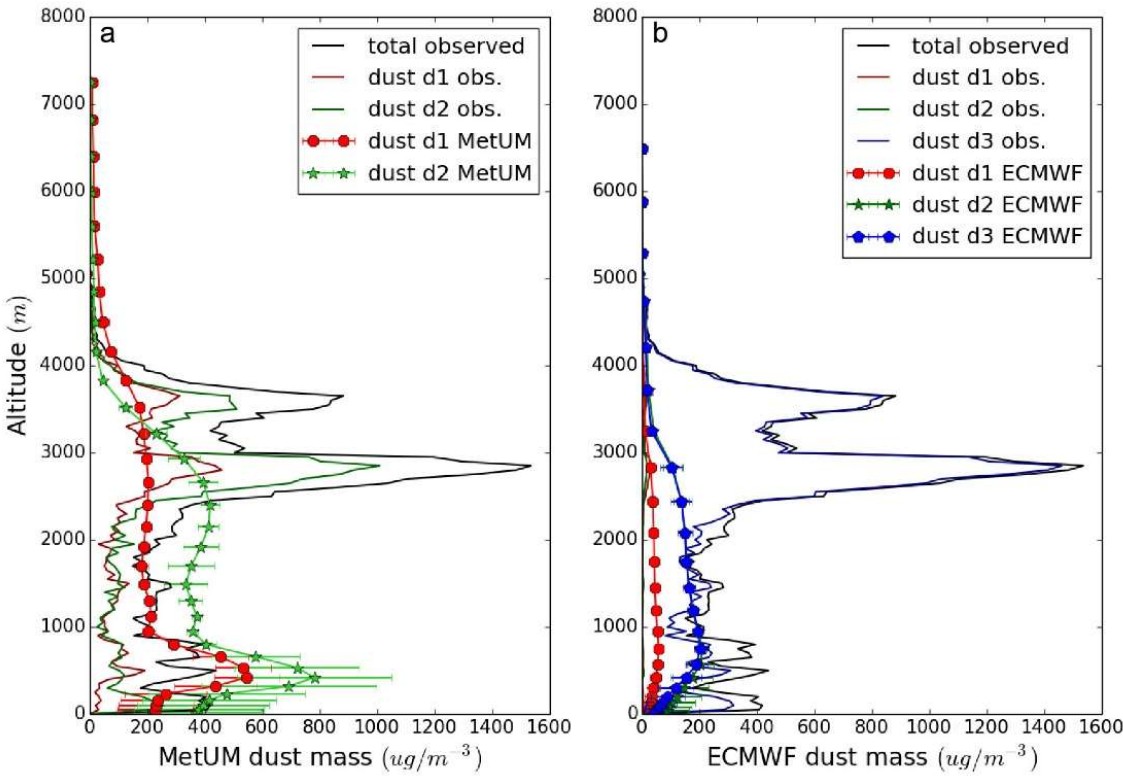

Figure 16: Case Study 4: B932, 20th August, P4. (a) Dust concentration measured by the in-situ instruments on the aircraft for two MetUM dust divisions 1 (red), and 2 (green), and the total dust concentration measured (black). The division 1 and 2 concentration from the model is shown in a lighter shade of red and green respectively, with markers and error bars showing the standard deviation. (b) The right hand plot shows the same thing but for the ECMWF CAMS size bins, with the measurements shown using lines, and the model values with lines and markers for divisions 1 (red), 2, (green), and 3 (blue).

P

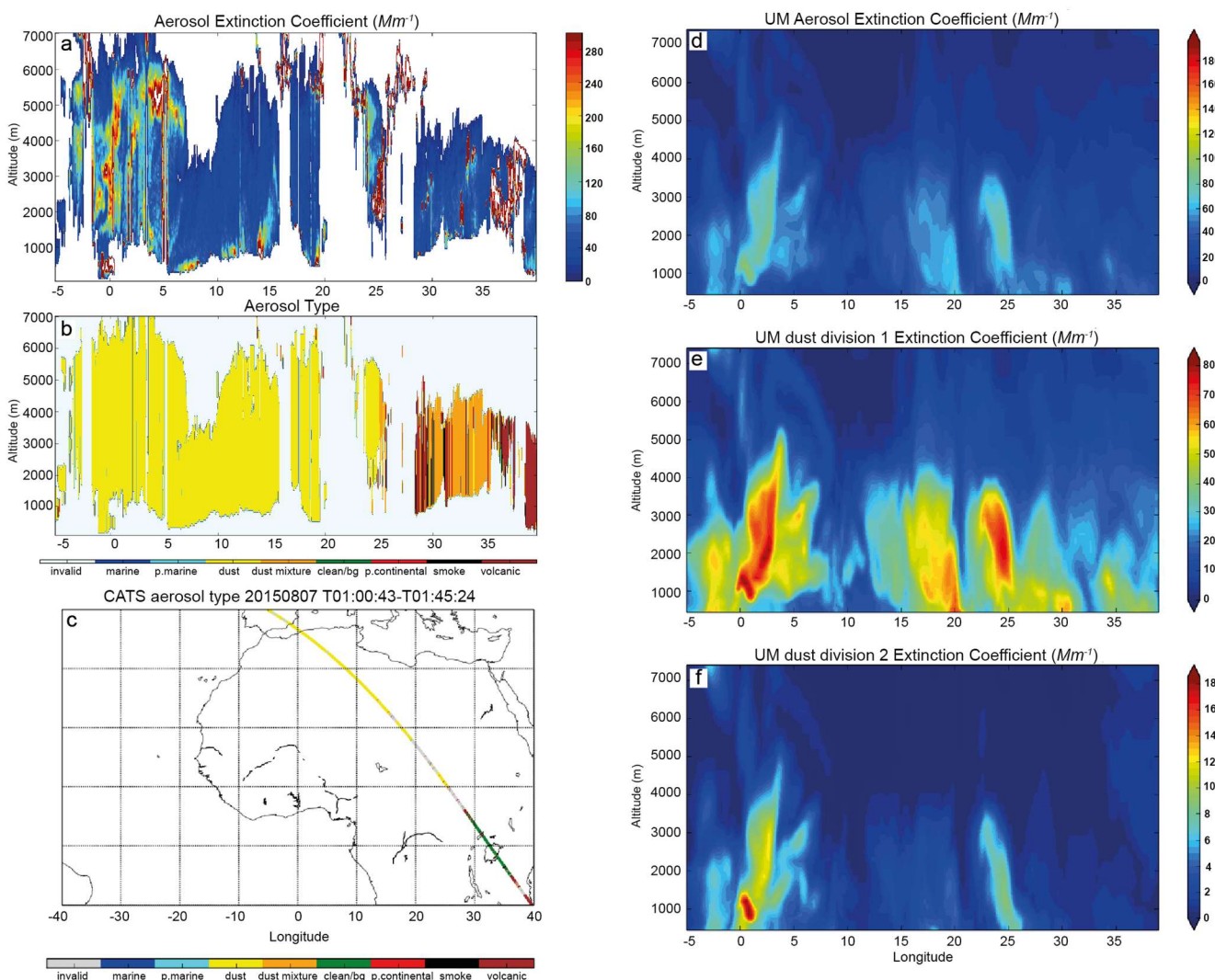

Figure 17: (a-c) CATS and (d-f) MetUM data for 00z on the 7ᵗʰ August, in the form of vertical cross–sections along the satellite track: (a) CATS extinction coefficient; (b) CATS feature type; (c) CATS overpass track; (d) MetUM total dust extinction coefficient; (e) MetUM d1 dust extinction coefficient; and (f) MetUM d2 dust extinction coefficient.

Q

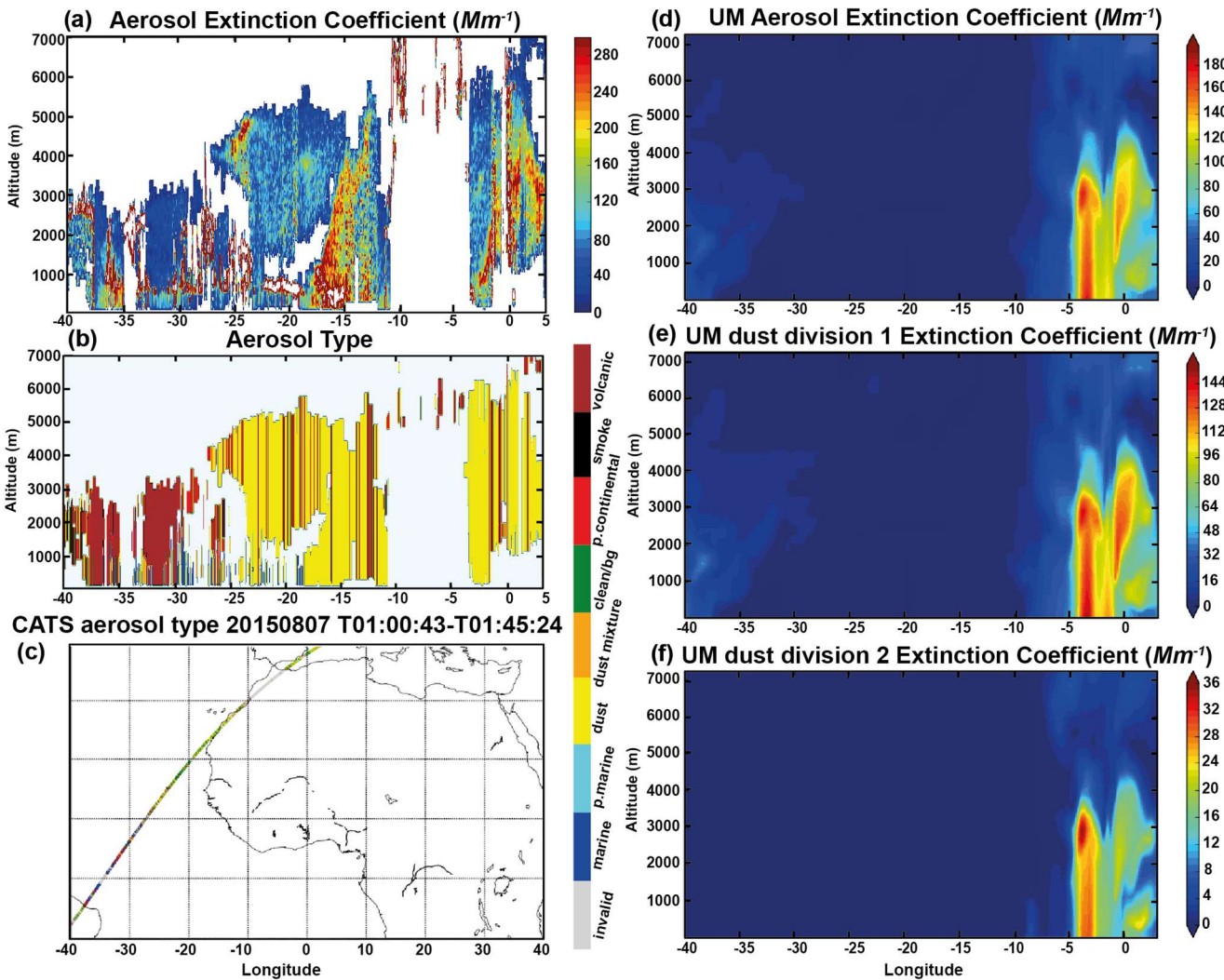

Figure 18: (a-c) CATS and (d-f) MetUM data for 18z on the 7th August, in the form of vertical cross–sections along the satellite track: (a) CATS extinction coefficient; (b) CATS feature type; (c) CATS overpass track; (d) MetUM total dust extinction coefficient; (e) MetUM d1 dust extinction coefficient; and (f) MetUM d2 dust extinction coefficient. Flight B920 is simultaneous to this satellite overpass near the Cape Verde islands.

R

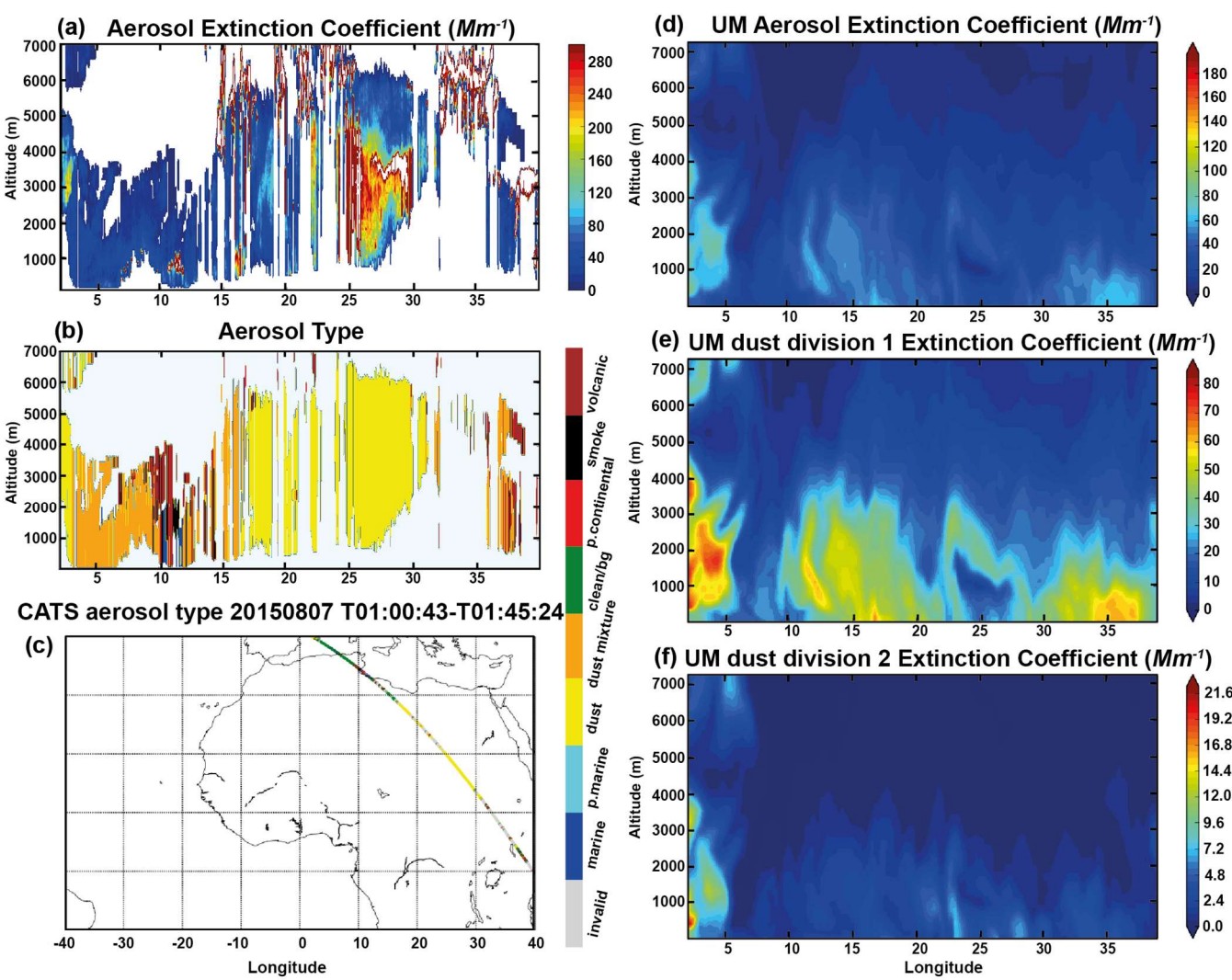

Figure 19: (a-c) CATS and (d-f) MetUM data for 00z on the 8th August, in the form of vertical cross–sections along the satellite track: (a) CATS aerosol extinction coefficient; (b) CATS feature type; (c) CATS overpass track; (d) MetUM total dust extinction coefficient; (e) MetUM d1 dust extinction coefficient; and (f) MetUM d2 dust extinction coefficient.

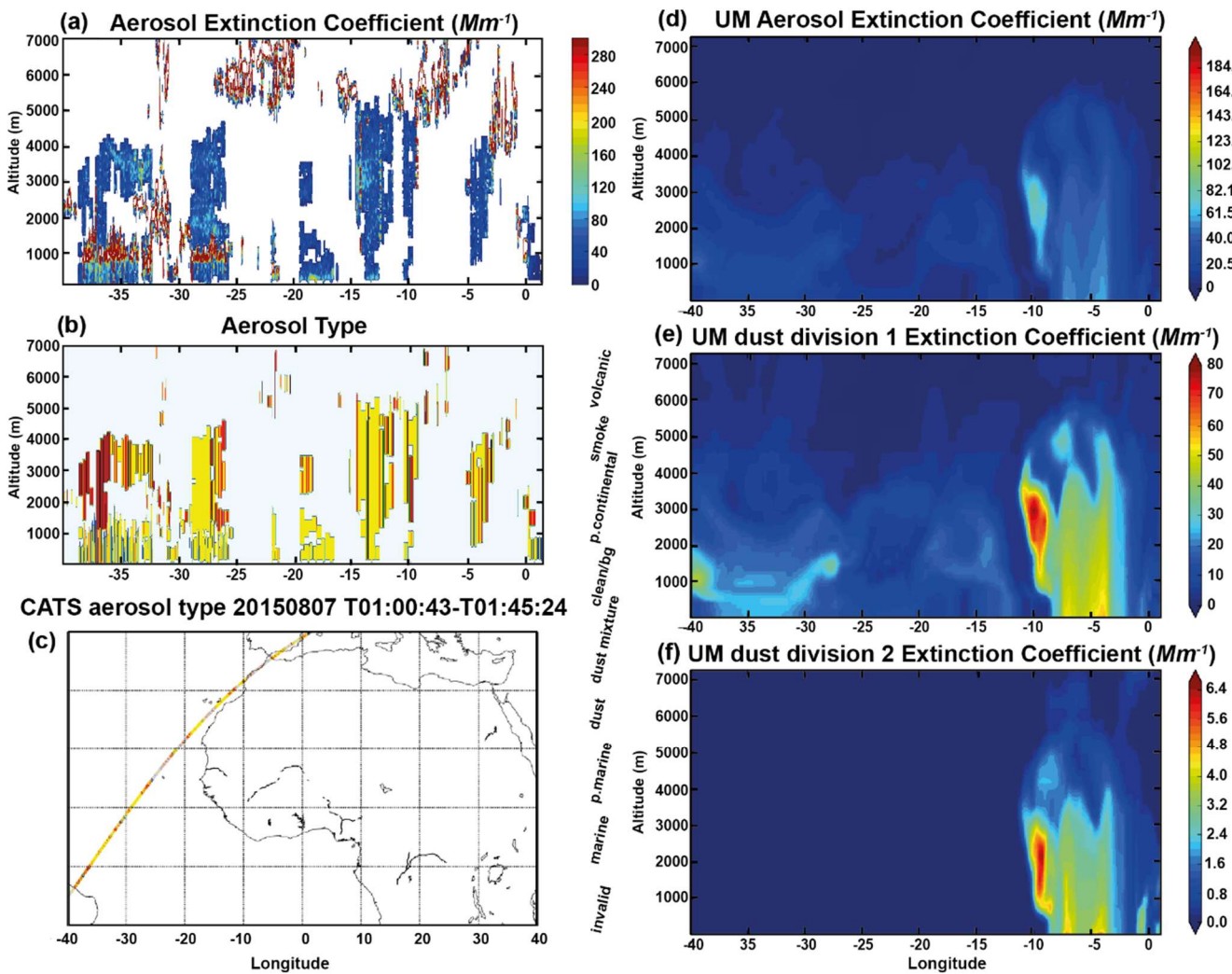

Figure 20: (a-c) CATS and (d-f) MetUM data for 00z on the 10th August, in the form of vertical cross–sections along the satellite track: (a) CATS aerosol extinction coefficient; (b) CATS feature type; (c) CATS overpass track; (d) MetUM total dust extinction coefficient; (e) MetUM d1 dust extinction coefficient; and (f) MetUM d2 dust extinction coefficient.

T

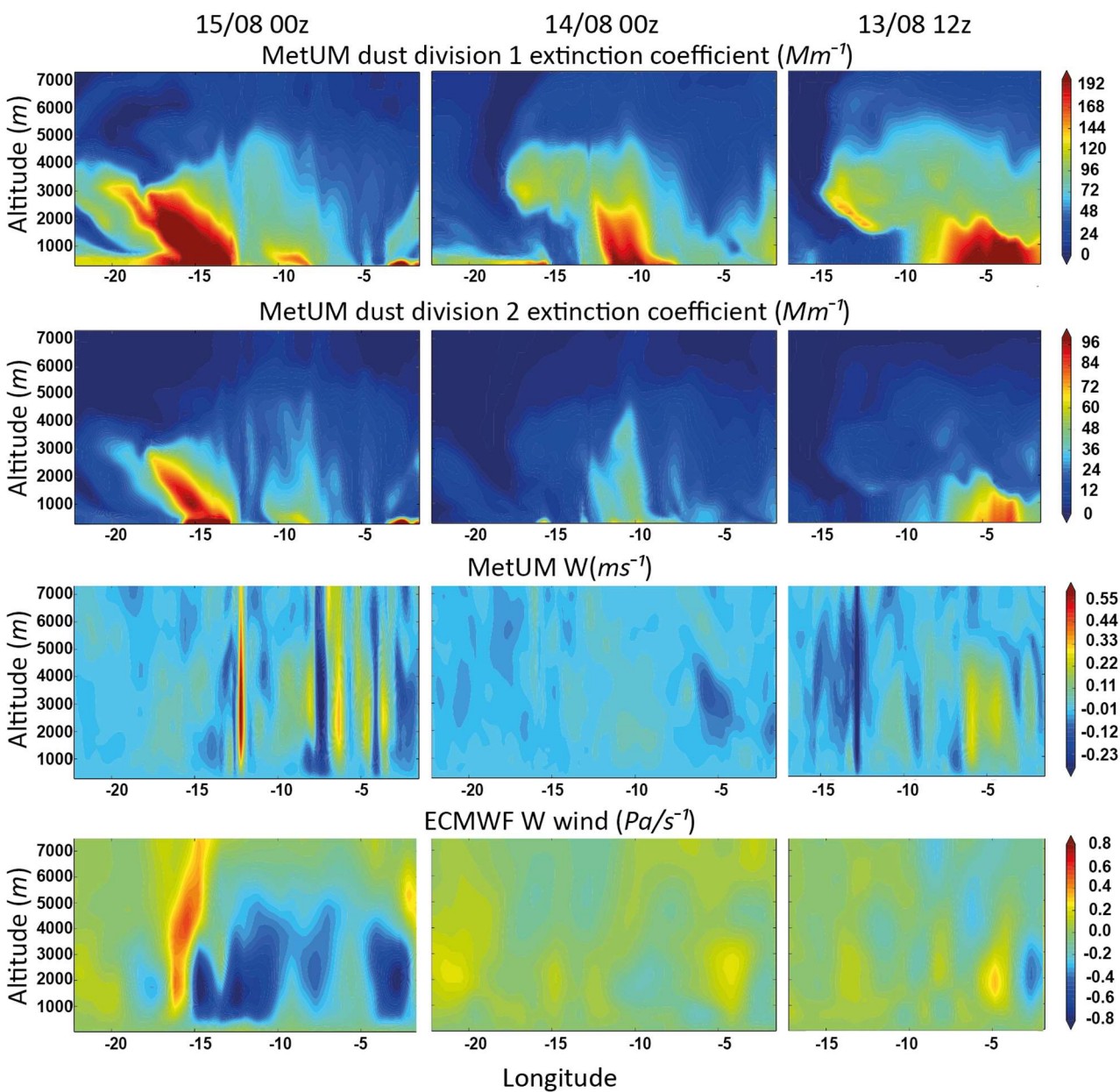

Figure 21: Contribution to the extinction coefficient by MetUM dust divisions d1 and d2 (top two rows), MetUM Westerly wind component, and ECMWF CAMS largescale wind. These cross-sections are extracted along the dust trajectory shown in fig 1, for case study 3 (flight B927, 15[th] August).

U