# Peer review of "Models transport Saharan dust too low in the atmosphere: a comparison of the MetUM and CAMS forecasts with observations"

_Atmospheric Chemistry and Physics, 2020_

## Referee Comment (RC1) · Anonymous Referee #2 · 14 May 2020

The paper "Models transport Saharan dust too low in the atmosphere compared to observations" presents and discusses the performance of two dust forecasts from MetUM and CAMS atmospheric models in comparison with in-situ measurements of dust size-distribution, airborne lidar derived optical properties and satellite based MODIS-Aqua and CATS-ISS observations obtained during August 2015 over the Eastern Atlantic Ocean, in the Saharan dust outflow vicinity, in the framework of the AER-D/ICE-D campaigns. The study, offering insight on dust transport, falls within the scope of ACP. The manuscript is well-written and well-structured, the presentation clear, the language fluent and the quality of the figures high. The authors have done a thorough job and the results support the conclusions. I recommend publication in ACP, however I recommend the following minor revisions before publication.

[Figure]

Comments:

1) As stated by the authors, MODIS AOD is assimilated in both MetUM and CAMS models. In MetUM MODIS C5 AOD is assimilated. Is this the same collection that is assimilated in CAMS? If not, how are the conclusions affected by the different assimilated collections? Furthermore, the comparison of the models is performed against MODIS C6.1. How do the different satellite datasets, the assimilation of different collections, and the non model-sattelite independence qualitative/quantitative affect the conclusions?

2) Regarding CATS, only two sentences are provided, in the "Satellite Data" Section. However, CATS is extensively used in the manuscript. The authors should extend the section with a proper description of the dataset, including information on the Version that is used (if not the latest version it is suggested to use V3.1), including in addition information on the Quality Assurance procedures that are followed prior to the comparison with the models and the FAAM airborne dataset.

3) Regarding the discussion of the comparisons made between MetUM, CAMS, FAAM lidar, CATS, and MODIS, the authors frequently remain to qualitative presentation of the results, without providing any quantitative values. For instance, the authors frequently use phrases like "very little wavelength dependence was noted", "there is virtually no difference", "agrees well", "are in agreement", "is broadly in agreement with", "under-predict the intensity", "is less than half", "the magnitude of the predicted extinction is similar, although with differences in the dust layering", without providing values. The entire manuscript should be revised accordingly.

4) In Table 1, statistical metrics are provided for the different AER-D flights, for MetUM, CAMS and FAAM lidar. It would be beneficial for the manuscript to include a flowchart showing the methodology of the comparison followed by the authors. The entire process can be summarized there along with the followed comparison methodology and requirements e.g. the spatial - temporal constraints, screening requirements, Quality

[Figure]

Assurance approach, etc. The information exists in the manuscript but I feel like it is scattered among the sections. Furthermore, I suggest the authors to provide the collocation criteria (both spatial and temporal), wavelengths, etc, since the datasets are very different. Finally, Table 1 should include more statistical metrics than the minimum, maximum and standard deviation.

5) Regarding references, the authors give proper credit to related work, especially in the introduction and the methodology sections. However, regarding the basic concept of the performance of dust models in dust transport and the main findings, I would suggest the authors to expand the discussion and the list of references in order to strengthen the manuscript and at the same in order to give credit to related studies, and additionally to discuss how the findings of the studies compare.

6) During August 2015, CATS operating on board the ISS was one of the two satellite-based operational lidar systems. Regarding ICE-D and the performed B920 ISS underflight, near the Cape Verde islands, the high collocated airborne and CATS lidar observations is a clear reason for the implementation of CATS. However, due to the broader study domain and the performed extended analysis in Section 4.3 "Comparison with the CATS spaceborne lidar", I am a bit surprised that the authors do not attempt to use a similar CALIOP-CALIPSO lidar approach. Dust retrieval is probably one of the best products from CALIPSO, even if CALIPSO reports only at 01:30 and 13:30 hrs local time, it should be useful to compare the MetUM and CAMS at those local times.

7) "The Cloud-Aerosol Transport System (CATS) onboard the International Space Station was a polarization sensitive backscatter lidar with higher detection sensitivity than CALIOP and superior ability to differentiate different aerosol types (Yorks et al., 2016)". The authors are kindly asked to check this statement. According to Table 1 of Yorks et al., 2016, although the MDB of CATS M7.2 1064nm is lower than CALIOP 1064nm during nighttime, this does not hold during daytime. Furthermore, this case study corresponds to a case study of cirrus clouds at 15km.

8) The quality of the Figures is high. Regarding the visualization of the extinction coefficient cross-section from CATS, it is suggested the authors to use a similar approach as done in the airborne lidar cross-sections, regarding missing calues (e.g. totally attenuated due to clouds, quality filtering, etc). In the way that the cross sections are provided in the present version of the manuscript, instead of missing values, values "zero" are assigned, leading to misinterpretations to the reader.

---

## Referee Comment (RC2) · Anonymous Referee #1 · 15 Jun 2020

The authors present a study with combined modeling, remote sensing and in-situ data and they point out certain deviations between the operational modeling simulations for dust transport and the actual observations. The performance of dust transport models and especially of those with lower resolutions is known to be problematic especially for the long range transport. This work identifies certain modeling elements that may be responsible for these discrepancies including the particle size distribution, optical properties calculations and prediction of wind fields. The study is clearly written and it can be an interesting addition to the relevant literature. I recommend publication with minor revisions. Specific comments are following below:

- Title: "Models transport Saharan dust too low in the atmosphere compared to observations". The title should change to something referring to MetUM and CAMS specif-

[Figure]

ically because it is not justified in the paper that all available models at all possible configurations transport the dust too low in the atmosphere.

- This work highlights an important consideration regarding the contribution of the smaller dust particles to the extinction coefficient and how this is used to compensate for the absent of large particles in dust models. The models are fine-tuned to represent the observed optical properties, which is sufficient for further direct effect studies (i.e. radiative transfer). However the indirect processes (e.g. IN/CCN activation) are mostly governed by particle concentration and thus it is not possible to obtain reasonable results for the indirect effect in the aforementioned models. This should be highlighted also in the abstract.

- [Line 25] "An analysis of the processes driving dust uplift in the models suggests that errors in the large scale wind". The large scale wind issue is not adequately supported by the results of this study.

- [Line 40] Please put the references in chronological order

- [Line 107] "….uses the 2-bin dust scheme, with a fine and accumulation mode bin (division 1 or d1, 0.2 - 4.0 $\mu$m diameter) and a coarse mode bin: (division 2 or d2, 4.0 - 20 $\mu$m diameter)", but in the paper you generally assume <1um as fine and > 1um as coarse. Please clarify how the MetUM bins fit into the general fine-coarse division in your study.

- [Line 225] "….  using the Numerical Atmospheric Modelling Environment (NAME) (Jones et al, 2007) and the Hybrid Single-Particle Lagrangian Integrated Trajectory model (HYSPLIT)". Please provide information on the meteorological data that you used to drive the Lagrangian simulations. Is it from MetUM and CAMS or from other models?

- [Line 310] "As already highlighted from Fig 2, the MetUM has the dust layer extending right down to the ocean surface. It is moreover dominated by the smaller size bin (d1,

0.2 – 4.0 $\mu$m diameter), in particular for the aerosol below 1km primarily. The concentration predicted by CAMS for this case is less than half of that in the MetUM, and however the magnitude of the predicted extinction is similar, although with differences in the dust layering". It would be useful for the reader to elaborate more on these concentration/optical products comparisons and possibly to explain also the methodology that you used to retrieve the extinction efficiencies.

- [Line 350] "The differences between models and observations could possibly be associated to the dust having been uplifted by a strong haboob, for which models may fail to capture the strength of the uplift (Roberts et al., 2018)." You could explain further here the reasons why these models do not capture the convective density currents e.g. due to their coarse resolution.

- [Line 356] "Compared to the very large differences between the measured and modelled dust concentration, the modelled extinction is much closer to the observations." Again this is an interesting highlight of this study and should be emphasized more and included in the abstract.

- [Line 485] "Previous studies which have looked at this issue more comprehensively do however suggest that there is an underprediction of wind fields in the models, which is also linked to coarse resolution modelling (eg. Chouza et al., 2016). Evan et al, (2016) showed that desert dust emission is to first order a function of wind speed, and it is against this quantity that models parametrise the dust source. Therefore, it seems reasonable that improving the wind speed in the models is a key part of getting the amount of dust uplift right." It is already well known that wind plays a key role in dust mobilization and transport. Please elaborate more on these wind considerations if you want to make a specific statement for your study as a simple reference in previous papers might not be adequate in this case.

---

## Author Comment (AC1) · 15 Jul 2020

We would like to thank the two reviewers for their very positive comments and for carefully reading our manuscript and formulating suggestions for improvement. We have followed the advice provided, and we believe that as a result the manuscript is now very much improved. Hereby we present a point by point response.

*Note: Line numbers in reviewers comments refer to the discussion paper, viewable online. Line numbers in our response to reviewers refer to the newly submitted version with track changes, reproduced at the end of this file.*

**Anonymous Referee #1**

The authors present a study with combined modeling, remote sensing and in-situ data and they point out certain deviations between the operational modeling simulations for dust transport and the actual observations. The performance of dust transport models and especially of those with lower resolutions is known to be problematic especially for the long range transport. This work identifies certain modeling elements that may be responsible for these discrepancies including the particle size distribution, optical properties calculations and prediction of wind fields. The study is clearly written and it can be an interesting addition to the relevant literature. I recommend publication with minor revisions. Specific comments are following below:

We thank the reviewer very much for appreciating this research and the way we chose to present it. We feel that the advice provided is very useful for the improvement of the manuscript.

- Title: "Models transport Saharan dust too low in the atmosphere compared to observations". The title should change to something referring to MetUM and CAMS specifically because it is not justified in the paper that all available models at all possible configurations transport the dust too low in the atmosphere.

We appreciate this comment, and we have modified the title accordingly to reflect the fact that the analysis has been carried out on those two models solely.

- This work highlights an important consideration regarding the contribution of the smaller dust particles to the extinction coefficient and how this is used to compensate for the absent of large particles in dust models. The models are fine-tuned to represent the observed optical properties, which is sufficient for further direct effect studies (i.e. radiative transfer). However the indirect processes (e.g. IN/CCN activation) are mostly governed by particle concentration and thus it is not possible to obtain reasonable results for the indirect effect in the aforementioned models. This should be highlighted also in the abstract.

This is a very good point. We have added the following in the abstract, which should address it: "We also found that both model forecasts underpredict coarse mode dust, and at times overpredict fine mode dust, but as they are fine-tuned to represent the observed optical depth, the fine mode is set

to compensate for the underestimation of the coarse mode. As aerosol-cloud interactions are dependent on particle numbers rather than on the optical properties, this behaviour is likely to affect their correct representation. This leads us to propose an augmentation of the set of aerosol observations available on a global scale for constraining models, with a better focus on the vertical distribution and on the particle size-distribution." (lines 23-28)

- [Line 25] "An analysis of the processes driving dust uplift in the models suggests that errors in the large scale wind". The large scale wind issue is not adequately supported by the results of this study.

We agree with the reviewer and we have therefore removed this sentence from the abstract.

- [Line 40] Please put the references in chronological order

We have now corrected this (line 42)

- [Line 107] ": : :.uses the 2-bin dust scheme, with a fine and accumulation mode bin (division 1 or d1, 0.2 - 4.0 m diameter) and a coarse mode bin: (division 2 or d2, 4.0 - 20 m diameter)", but in the paper you generally assume <1um as fine and > 1um as coarse. Please clarify how the MetUM bins fit into the general fine-coarse division in your study.

We agree with the reviewer that this terminology may have led to confusion. We have therefore reworded the manuscript by describing the model size bins in terms of particle sizes, but without any reference to the fine, accumulation, and coarse mode. (lines 112-113)

- [Line 225] ": : :. using the Numerical Atmospheric Modelling Environment (NAME) (Jones et al, 2007) and the Hybrid Single-Particle Lagrangian Integrated Trajectory model (HYSPLIT)". Please provide information on the meteorological data that you used to drive the Lagrangian simulations. Is it from MetUM and CAMS or from other models?

NAME was driven from MetUM data, and HYSPLIT from GDAS. We have now added this information in the paper. (lines 255-258)

- [Line 310] "As already highlighted from Fig 2, the MetUM has the dust layer extending right down to the ocean surface. It is moreover dominated by the smaller size bin (d1, 0.2 – 4.0 um diameter), in particular for the aerosol below 1km primarily. The concentration predicted by CAMS for this case is less than half of that in the MetUM, and however the magnitude of the predicted extinction is similar, although with differences in the dust layering". It would be useful for the reader to elaborate more on these concentration/optical products comparisons and possibly to explain also the methodology that you used to retrieve the extinction efficiencies.

We have now added an explanatory sentence (lines 343-345) and we have added the methodology used to retrieve the extinction efficiencies in each model at the end of sections 2.1 and 2.2 (lines 120-124 and 145-147).

- [Line 350] "The differences between models and observations could possibly be associated to the dust having been uplifted by a strong haboob, for which models may fail to capture the strength of the uplift (Roberts et al., 2018)." You could explain further here the reasons why these models do not capture the convective density currents e.g. due to their coarse resolution.

We have now added the following: "The differences between models and observations could possibly be associated to the dust having been uplifted by a strong haboob, for which models, running with the resolution and convection parameterisation required for global coverage, are unlikely to represent in a way that gives the strength of the uplift (Marsham et al. 2013b, Birch et al. 2014, Roberts et al., 2018). In particular, we note that the convection parametrisation has no specific representation of surface gusts due to downdrafts (main contributors to dust uplift) and that it isn't currently coupled to the dust scheme." (lines 385-390)

- [Line 356] "Compared to the very large differences between the measured and modelled dust concentration, the modelled extinction is much closer to the observations." Again this is an interesting highlight of this study and should be emphasized more and included in the abstract.

We have added the following to the abstract to highlight this point: "We also found that both model forecasts underpredict coarse mode dust, and at times overpredict fine mode dust, but as they are fine-tuned to represent the observed optical depth, the fine mode is set to compensate for the underestimation of the coarse mode." (lines 23-25)

- [Line 485] "Previous studies which have looked at this issue more comprehensively do however suggest that there is an underprediction of wind fields in the models, which is also linked to coarse resolution modelling (eg. Chouza et al., 2016). Evan et al, (2016) showed that desert dust emission is to first order a function of wind speed, and it is against this quantity that models parametrise the dust source. Therefore, it seems reasonable that improving the wind speed in the models is a key part of getting the amount of dust uplift right." It is already well known that wind plays a key role in dust mobilization and transport. Please elaborate more on these wind considerations if you want to make a specific statement for your study as a simple reference in previous papers might not be adequate in this case.

We agree with the reviewer that we cannot draw conclusions from our study, we have therefore removed the corresponding statement from the abstract, and we have reworded this sentence as follows: "This combined with our observations of an increase in largescale wind velocity at the time of dust uplift suggest that further investigation into the role of wind speed in the models would be helpful as a key part of getting the amount of dust uplift right. " (lines 542-544)

**Anonymous Referee #2**

The paper "Models transport Saharan dust too low in the atmosphere compared to observations" presents and discusses the performance of two dust forecasts from MetUM and CAMS atmospheric models in comparison with in-situ measurements of dust size distribution, airborne lidar derived optical properties and satellite based MODIS-Aqua and CATS-ISS observations obtained during August 2015 over the Eastern Atlantic Ocean, in the Saharan dust outflow vicinity, in the framework of the AER-D/ICE-D campaigns. The study, offering insight on dust transport, falls within the scope of ACP. The manuscript is well-written and well-structured, the presentation clear, the language fluent and the quality of the figures high. The authors have done a thorough job and the results support the conclusions. I recommend publication in ACP, however I recommend the following minor revisions before publication.

We thank the reviewer for the constructive advice and for his/her appreciation of our article. In the following we shall address his/her comments, which we think have led to an improvement of the paper.

1) As stated by the authors, MODIS AOD is assimilated in both MetUM and CAMS models. In MetUM MODIS C5 AOD is assimilated. Is this the same collection that is assimilated in CAMS? If not, how are the conclusions affected by the different assimilated collections? Furthermore, the comparison of the models is performed against MODIS C6.1. How do the different satellite datasets, the assimilation of different collections, and the non model-sattelite independence qualitative/quantitative affect the conclusions?

Good point, thanks. In 2015, the MetUM actually assimilated C5.1 (correction made, line 114) whereas CAMS assimilated C5 (now specified on line 128), whereas the comparisons are with the later product C6.1. In section 3.4 we have added a paragraph outlining the differences between these data versions: "The differences between the collection 5 (used in both models for operational assimilation in August 2015) and the subsequently released collection 6 is treated in detail in the above referenced papers. Generally speaking, with the collection 6 update, the Deep Blue product was extended to vegetated surfaces, and improvements to the aerosol type classification and quality assurance were introduced for both the Dark Target and Deep Blue products. Comparisons performed by the authors suggest that, generally speaking, the collection 6 AOD values are marginally higher in the dust source regions (e.g. Western Africa and Middle East). The differences between the MODIS collections represent a major improvement to the MODIS product, but we do not expect them to substantially affect the conclusions drawn in this paper." (lines 242-249)

2) Regarding CATS, only two sentences are provided, in the "Satellite Data" Section. However, CATS is extensively used in the manuscript. The authors should extend the section with a proper description of the dataset, including information on the Version that is used (if not the latest version it is suggested to use V3.1), including in addition information on the Quality Assurance procedures that are followed prior to the comparison with the models and the FAAM airborne dataset.

We have now provide this information in section 3.4 as follows: "CATS operated for 33 months (10 February 2015 to 29 October 2017), primarily in an operating mode that was limited to the 1064 nm wavelength due to issues with stabilizing the frequency of laser 2 (Yorks et a., 2016). The CATS Level

1 data product includes 1064 nm attenuated total backscatter (ATB) and linear volume depolarization ratio measurements. Yorks et al. (2016) provides an overview of the CATS L1 data products and processing algorithms and a comparison with airborne data. Pauly et al. (2019) found that the CATS 1064 nm ATB has a low bias of up to 7% in aerosol layers compared to airborne and ground based lidars due primarily to CATS calibration uncertainties. The CATS extinction coefficient profiles have a 5 km horizontal resolution (along-track) and 60 m vertical resolution. Lee et al. (2019) showed that CATS extinction profiles compared favorably with CALIPSO, with differences due to the aforementioned ATB bias and differences in parameterized extinction-to-backscatter ratios. This paper utilizes the vertical profiles of 1064 nm aerosol extinction coefficient in the CATS Level 2 (L2) Version 3-01 5km Profile products derived from the L1 attenuated total backscatter data. For this study, the data were filtered by the 'cloud' and 'invalid' flags, thus showing only the aerosol data points. The aerosol subtype (plotted together with extinction coefficient) indicates that most of the aerosol of interest here is in fact classified as dust and dust mixtures in the CATS L2 dataset." (lines 224-237)

3) Regarding the discussion of the comparisons made between MetUM, CAMS, FAAM lidar, CATS, and MODIS, the authors frequently remain to qualitative presentation of the results, without providing any quantitative values. For instance, the authors frequently use phrases like "very little wavelength dependence was noted", "there is virtually no difference", "agrees well", "are in agreement", "is broadly in agreement with", "underpredict the intensity", "is less than half", "the magnitude of the predicted extinction is similar, although with differences in the dust layering", without providing values. The entire manuscript should be revised accordingly.

We have now reworded our qualitative statements throughout section 4, in order to offer a more quantitative comparison of datasets. Please see the highlighted changes in the manuscript appended to this response file.

4) In Table 1, statistical metrics are provided for the different AER-D flights, for MetUM, CAMS and FAAM lidar. It would be beneficial for the manuscript to include a flowchart showing the methodology of the comparison followed by the authors. The entire process can be summarized there along with the followed comparison methodology and requirements e.g. the spatial - temporal constraints, screening requirements, Quality Assurance approach, etc. The information exists in the manuscript but I feel like it is scattered among the sections. Furthermore, I suggest the authors to provide the collocation criteria (both spatial and temporal), wavelengths, etc, since the datasets are very different.

Many thanks for this suggestion. We have now added a flow chart summarising the datasets used for AOD, dust mass and extinction, and the methods applied for the comparison. We believe that this addition will make the methodology clearer and the paper easier to read. (line 305) and Figure 2

Finally, Table 1 should include more statistical metrics than the minimum, maximum and standard deviation.

Percentiles could in principle be added but to our opinion they would just add complexity, whereas the difference between the model main aerosol properties and the observed ones are already well captured with the mean and the maximum. It is to note that these distributions are obtained over large distances (e.g. B923 over 1,500 km) and over the full vertical extent from the airplane to the

surface, hence it is expected for them to be very irregular statistical functions. We feel that it is better for the reader to concentrate on the figures, to get an idea of the spatial distribution.

5) Regarding references, the authors give proper credit to related work, especially in the introduction and the methodology sections. However, regarding the basic concept of the performance of dust models in dust transport and the main findings, I would suggest the authors to expand the discussion and the list of references in order to strengthen the manuscript and at the same in order to give credit to related studies, and additionally to discuss how the findings of the studies compare.

We thank the reviewer for this comment, however we point out that some of the literature was already cited in the previous version of the introduction (Heintzenberg, 2009; Ansmann et al., 2011; Kanitz et al.; 2014, Ryder et al.; 2015, Groβ et al.; 2015; Chouza et al., 2016; Ansmann et al., 2017; Evan, 2018) and of the conclusions (Kim et al, 2014; Mona et al, 2014; Binietoglou et al., 2015; and Ansmann et al, 2017).

We have now expanded the conclusions and we have added reference to more previous works, citing Roberts et al (2018) at lines 574-575; Konsta et al (2018) at lines 580-582; and Adebiyi and Kok (2020), Huneeus et al (2011) and Hoshyaripour et al (2019) at lines 593-599. Should the reviewer feel that more literature citations are missing, we would however kindly ask them to indicate the papers that he or she regards as key references in this context.

6) During August 2015, CATS operating on board the ISS was one of the two satellite-based operational lidar systems. Regarding ICE-D and the performed B920 ISS underflight, near the Cape Verde islands, the high collocated airborne and CATS lidar observations is a clear reason for the implementation of CATS. However, due to the broader study domain and the performed extended analysis in Section 4.3 "Comparison with the CATS spaceborne lidar", I am a bit surprised that the authors do not attempt to use a similar CALIOP-CALIPSO lidar approach. Dust retrieval is probably one of the best products from CALIPSO, even if CALIPSO reports only at 01:30 and 13:30 hrs local time, it should be useful to compare the MetUM and CAMS at those local times.

We agree with the reviewer, however it is beyond the scope of this research to further expand the datasets used to CALIPSO. The article is already long and the analysis has taken us quite a bit of time. We do not rule out, however, doing a similar study again in the future, using more datasets, improved versions of the models, and a data assimilation framework that can include more observations.

7) "The Cloud-Aerosol Transport System (CATS) onboard the International Space Station was a polarization sensitive backscatter lidar with higher detection sensitivity than CALIOP and superior ability to differentiate different aerosol types (Yorks et al., 2016)". The authors are kindly asked to check this statement. According to Table 1 of Yorks et al., 2016, although the MDB of CATS M7.2 1064nm is lower than CALIOP 1064nm during nighttime, this does not hold during daytime. Furthermore, this case study corresponds to a case study of cirrus clouds at 15km.

It is beyond our purposes to compare CATS to CALIPSO and we have therefore modified the sentence as follows: "The Cloud-Aerosol Transport System (CATS) onboard the International Space Station was a polarisation sensitive backscatter lidar with good detection sensitivity and ability to differentiate different aerosol types" (line 84).

8) The quality of the Figures is high. Regarding the visualization of the extinction coefficient cross-section from CATS, it is suggested the authors to use a similar approach as done in the airborne lidar cross-sections, regarding missing calues (e.g. totally attenuated due to clouds, quality filtering, etc). In the way that the cross sections are provided in the present version of the manuscript, instead of missing values, values "zero" are assigned, leading to misinterpretations to the reader.

This is a good point and we have addressed it. See new figures 17-20.

[revised manuscript text omitted]

---

## Editor Decision (ED1)

**Models transport Saharan dust too low in the atmosphere: a comparison of the MetUM and CAMS forecasts with observations**

Debbie O'Sullivan1, Franco Marenco1, Claire L. Ryder2, Yaswant Pradhan1, Zak Kipling3, Ben Johnson1, Angela Benedetti3, Melissa Brooks1, Matthew McGill4, John Yorks4, Patrick Selmer4

5

1Met Office, Exeter, EX1 3PB, UK

2Department of Meteorology, University of Reading, RG6 6BB, UK
3European Centre for Medium-Range Weather Forecasts, Reading, RG2 9AX, UK
4NASA Goddard Space Flight Center, Greenbelt, MD 20771, USA

10 Correspondence to: Franco Marenco (franco.marenco@metoffice.gov.uk)

Abstract. We investigate the dust forecasts from two operational global atmospheric models in comparison with in-situ and remote sensing measurements obtained during the AERosol properties – Dust (AER-D) field campaign. Airborne elastic backscatter lidar measurements were performed on-board the Facility for Airborne Atmospheric Measurements during August 2015 over the Eastern Atlantic, and they permitted to characterize the dust vertical

- 15 distribution in detail, offering insights on transport from the Sahara. They were complemented with airborne insitu measurements of dust size-distribution and optical properties, and datasets from the Cloud-Aerosol Transport System spaceborne lidar (CATS) and the Moderate Resolution Imaging Spectroradiometer (MODIS). We compare the airborne and spaceborne datasets to operational predictions obtained from the Met Office Unified Model (MetUM) and the Copernicus Atmosphere Monitoring Service (CAMS). The dust aerosol optical depth predictions
- 20 from the models are generally in agreement with the observations, but display a low bias. However, the predicted vertical distribution places the dust lower in the atmosphere than highlighted in our observations. This is particularly noticeable for the MetUM, which does not transport coarse dust high enough in the atmosphere, nor far enough away from source. We also found that both model forecasts underpredict coarse mode dust, and at times overpredict fine mode dust, but as they are fine-tuned to represent the observed optical depth, the fine mode is set
- 25 to compensate for the underestimation of the coarse mode. As aerosol-cloud interactions are dependent on particle numbers rather than on the optical properties, this behaviour is likely to affect their correct representation. This leads us to propose an augmentation of the set of aerosol observations available on a global scale for constraining models, with a better focus on the vertical distribution and on the particle size-distribution. Mineral dust is a major component of the climate system, therefore it is important to work towards improving how models reproduce its
- 30 properties and transport mechanisms.

**1** Introduction**

Mineral dust is an important component of the Earth system (Forster et al., 2007, Haywood and Boucher, 2010, Knippertz and Todd, 2012), and it affects the scattering and absorption of solar and infrared radiation, as well as cloud microphysics. The Saharan desert is the main source of mineral dust (Washington et al., 2003; Shao et al., 2011) and here a life distant and here are between the search are the search and the data and the data and the search are the search and the search are the search are the search and the search are the search

[revised manuscript text omitted]
      | 14:29:28 | 15.56 | 22.98 | ECMWF                                              | 55   | 38    | 126  |
|                |         | to       | to    | to    | MetUM                                              | 58   | 41    | 177  |
|                |         | 15:00:25 | 17.54 | 21.40 | Lidar                                              | 57   | 47    | 329  |
|                |         | 17:29:11 | 15.54 | 22.99 | ECMWF                                              | 55   | 38    | 126  |
|                | R6      | to       | to    | to    | MetUM                                              | 58   | 41    | 177  |
|                |         | 18:00:18 | 17.54 | 21.40 | Lidar                                              | 56   | 40    | 212  |
| B923
12 Aug | R1      | 09:18:08 | 16.03 | 22.97 | ECMWF                                              | 140  | 120   | 490  |
|                |         | to       | to    | to    | MetUM                                              | 90   | 120   | 720  |
|                |         | 11:51:28 | 27.30 | 13.82 | Lidar                                              | 110  | 130   | 1130 |
| B924
12 Aug | R3-R4   | 15:11:57 | 23.26 | 18.81 | ECMWF                                              | 169  | 97    | 485  |
|                |         | to       | to    | to    | MetUM                                              | 51   | 27    | 205  |
|                |         | 16:04:43 | 24.49 | 17.81 | Lidar                                              | 180  | 180   | 1260 |
|                | R6-R7   | 16:48:29 | 16.44 | 23.03 | ECMWF                                              | 107  | 84    | 443  |
|                |         | to       | to    | to    | MetUM                                              | 46   | 35    | 169  |
|                |         | 18:33:01 | 24.08 | 18.18 | Lidar                                              | 60   | 100   | 1150 |
| B927
15 Aug | R1      | 13:59:06 | 11.42 | 24.55 | ECMWF                                              | 81   | 60    | 332  |
|                |         | to       | to    | to    | MetUM                                              | 54   | 41    | 159  |
|                |         | 14:46:00 | 15.05 | 23.37 | Lidar                                              | 78   | 96    | 372  |
| B932
20 Aug | R1      | 09:52:41 | 17.72 | 21.19 | ECMWF                                              | 140  | 120   | 500  |
|                |         | to       | to    | to    | MetUM                                              | 140  | 130   | 620  |
|                |         | 10:35:23 | 20.67 | 18.93 | Lidar                                              | 76   | 81    | 395  |

950Table 1: Summary of the high-level sections from each of the flights used here. Flight sections are
labelled with the letter R (runs), see text. All times UTC.

| Flight         | Flight section | Time                       | Lat N                | Lon W                | Altitude
AMSL (km) |
|----------------|----------------|----------------------------|----------------------|----------------------|-----------------------|
| B920
7 Aug  | P1             | 14:03:33
to
14:25:21 | 14.94
to
15.76 | 22.78
to
23.48 | 0.1 to 6.5            |
|                | P2             | 15:02:59
to
15:24:05 | 16.26
to
17.43 | 21.37
to
22.38 | 0.1 to 6.5            |
|                | Ρ7             | 17:08:08
to
17:27:50 | 17.34
to
17.94 | 21.00
to
21.53 | 0.1 to 6.5            |
| B923
12 Aug | P1             | 11:51:28
to
12:09:28 | 27.30
to
28.44 | 13.71
to
13.87 | 0.1 to 6.9            |
| B932
20 Aug | P4             | 10:37:23
to
11:01:22 | 20.01
to
20.30 | 18.88
to
20.22 | 0.1 to 6.5            |

Table 2: Summary of the aircraft profiles from each of the flights used here. Flight sections are labelled with the letter P (profiles), see text. All times UTC.

---

## Author Response (AR2)

Dear authors,

thank you very much for your revision work that substantially improved the paper, as also acknowledged by the two Referees. There are only two points that according to Referee #2 have to be mentioned for the sake of completeness. I kindly ask you to take these points into account in order to finish with the revision and proceed with the publication of the paper.

Best regards

We would like to thank the editor and the two reviewers for their very positive comments and for carefully reading our manuscript and formulating suggestions for improvement. We have followed the additional advice provided by Referee #2, and we believe that the paper is now mature for publication.

**Anonymous Referee #1**

For final publication, the manuscript should be **accepted as is**.

We are persuaded now to have come with a very good article, thanks amongst others to the reviewer's advice in the first round of reviews. Thanks for appreciating the way in which we have dealt with the advice provided!

**Anonymous Referee #2**

The paper deserves publication to ACP in its current form.

There are two points though that have to be mentioned for the sake of completeness:

1. Dust transport models fine-tune to AOD observations as it is well-mentioned in the text. Thus, the conversion of concentrations to extinction values are critical and have to be improved (also mentioned). What is not mentioned is the role of particle shape in these conversions, something that can be acknowledged as a factor that should be taken into account in future (e.g. for conversions but also for improvements on drag coefficient calculations that would affect deposition and transport dynamics in general).

2. It should be also mentioned that dust microphysics and consequent radiative properties such as SSA and assymetry parameter alter heating rates and atmospheric thermodynamics, affecting also transport. RTMs within dust models should follow advances and integrate new findings on dust microphysics. For example, large particles are removed more efficiently, affecting size distribution during transport along with SSA and assymetry parameter of the samples, something that is not taken into account in RTMs.

We thank very much the reviewer for reminding these two important points. We have now added a paragraph at lines 638-646 that covers both of them.

---

## Author Response (AR3)

We have now made the corrections indicated by the editor (see below). Many thanks.

**Corrections:**

630     Additionally, some processes may deserve better attention, as studies suggest that they could increase the lifetime of coarse and giant particles beyond what is predicted for gravitational settling: e.g. turbulence within the Saharan Air Layer, particle electrification, and the role of convective systems (Van Der Does et al, 2018). The optimum balance between these processes is still to be understood, as is the correct estimation of emission intensity. The dust observable properties, in terms of the aerosol optical depth, the particle sizes,
635     the spatial distribution, and the vertical distribution, are determined by these processes. The combination of all these properties determines the impact of dust on the climate system, hence the importance of understanding these processes better (see e.g. Kok et al, 2017).

Two more points that need attention are the particle shape and effect of dust on the radiation field. If dust particles are assumed spherical, many computations are easier, however it is well-known that dust particles
640     are very irregular. The mass-to-extinction conversion and the drag coefficient calculations (which affects deposition and transport) are directly affected by particle shape. Moreover. dust microphysics and consequent radiative properties such as single-scattering albedo and asymmetry parameter do alter the computations of atmospheric radiation due to dust. In turns, this affects the heating rates of atmospheric layers, the atmospheric thermodynamics, the convective motions, and the wind fields which result in dust transport
645     patterns. An improvement of the radiative transfer models within dust models is therefore suggested, to integrate the latest understanding of dust microphysics.

[Figure]

**Corrected text:**

Two more points that need attention are the particle shape and effect of dust on the radiation field, atmospheric heating rates and thermodynamics and the dust transport itself. If dust particles are assumed
640     spherical in the dust transport models, many computations are easier, however it is well-known that dust particles are very irregular. The mass-to-extinction conversion and the drag coefficient calculations (which affect deposition and transport) are directly affected by particle shape. Moreover. dust microphysics and consequent radiative properties such as single-scattering albedo and asymmetry parameter do alter the computations of atmospheric radiation due to dust. In turn, this affects the heating rates of atmospheric layers,
645     the atmospheric thermodynamics, the convective motions, and the wind fields, which result in possible modifications of the dust transport patterns. An improvement of the radiative transfer models within dust models is therefore suggested, to integrate the latest understanding of dust microphysics.